# Nucleus accumbens dopamine release reflects Bayesian inference during instrumental learning

**Albert J. Qü** [1,2*], **Lung-Hao Tai**[3], **Christopher D. Hall**[4], **Emilie M. Tu** [5], **Maria K. Eckstein** [6], **Karyna Mishchanchuk**[4,7], **Wan Chen Lin**[3], **Juliana B. Chase**[1], **Andrew F. MacAskill**[7], **Anne G. E. Collins**[1,3], **Samuel J. Gershman**[8,9], **Linda Wilbrecht**[1,3]

1 Department of Psychology, University of California, Berkeley, Berkeley, California, United States of America, 2 Center for Computational Biology, University of California, Berkeley, Berkeley, California, United States of America, 3 Helen Wills Neuroscience Institute, University of California, Berkeley, Berkeley, California, United States of America, 4 Sainsbury Welcome Centre for Neural Circuits and Behaviour, University College London, London, United Kingdom, 5 Department of Neurobiology, Harvard Medical School, Boston, Massachusetts, United States of America, 6 Google DeepMind, London, United Kingdom, 7 Department of Neuroscience, Physiology and Pharmacology, University College London, London, United Kingdom, 8 Department of Psychology and Center for Brain Science, Harvard University, Cambridge, Massachusetts, United States of America, 9 Center for Brains, Minds, and Machines, Massachusetts Institute of Technology, Cambridge, Massachusetts, United States of America

* albert_qu@berkeley.edu

**Data availability statement:** The authors confirm that all data underlying the findings are

## Abstract

Dopamine release in the nucleus accumbens has been hypothesized to signal the difference between observed and predicted reward, known as reward prediction error, suggesting a biological implementation for reinforcement learning. Rigorous tests of this hypothesis require assumptions about how the brain maps sensory signals to reward predictions, yet this mapping is still poorly understood. In particular, the mapping is nontrivial when sensory signals provide ambiguous information about the hidden state of the environment. Previous work using classical conditioning tasks has suggested that reward predictions are generated conditional on probabilistic beliefs about the hidden state, such that dopamine implicitly reflects these beliefs. Here we test this hypothesis in the context of an instrumental task (a two-armed bandit), where the hidden state switches stochastically. We measured choice behavior and recorded dLight signals that reflect dopamine release in the nucleus accumbens core. Model comparison among a wide set of cognitive models based on the behavioral data favored models that used Bayesian updating of probabilistic beliefs. These same models also quantitatively matched mesolimbic dLight measurements better than non-Bayesian alternatives. We conclude that probabilistic belief computation contributes to instrumental task performance in mice and is reflected in mesolimbic dopamine signaling.

fully available without restriction. All dLight fiber photometry recordings and mouse behavior data generated in this paper are deposited in Dandi in NWB format (ID: 001340). https://dandiarchive.org/dandiset/001340?search=001340&pos=1. Modeling code is available publicly via github: https://github.com/Wilbrecht-Lab/cogmodels.

**Funding:** This work was financially supported by the National Institutes of Health (grant number U19NS113201 to L.W. and S.J.G.) and the Air Force Office of Scientific Research (grant number FA9550-20-1-0413 to S.J.G.) to cover logistical, experimental and personnel cost. The funders had no role in study design, data collection and analysis, decision to publish, or preparation of the manuscript.

**Competing interests:** The authors have declared that no competing interests exist.

## Author summary

We investigated how mice adapt to changes in the hidden reward structure in an instrumental task. We found evidence that models using Bayesian inference provided a better account of both behavioral data and dopamine signals in the Nucleus Accumbens for mice compared to standard Reinforcement Learning (RL) models, even those with sophisticated features such as counterfactuals, forgetting, and dynamic learning rate update. Moreover, we discovered that it is biologically plausible that mice employed a hybrid computational process that combined RL and Bayesian inference. Under this hypothesis, Bayesian inference is used to compute beliefs across hidden states, each of which contains its own action value map. Dopamine signals reward prediction errors that update these action values conditioned on these belief states. This work establishes a role for Bayesian inference in neural computation and behavior in a value-based decision making task, and helps to confirm a new understanding of the role of dopamine in learning.

## Introduction

Reinforcement learning (RL) algorithms hypothesize that, in value-based decision making tasks, animals maintain an action value map and update it using reward prediction errors (RPE), the difference between the observed and predicted reward associated with the chosen action [1]. The growing family of RL models proved to be remarkably successful in explaining trial-by-trial changes in behavior [2–4] and the responses of midbrain dopamine neurons in rodents and primates [5–7]. However, these standard RL (SRL) models may not adequately account for behavioral and neural data when immediate sensory observations alone are insufficient to guide optimal behavior, for example, when the underlying reward distribution changes over time due to the existence of a hidden state [8–10]. To solve such tasks, evidence suggests that the brain may represent a "belief state" (a probability distribution over hidden states), updated by Bayesian inference [10–15], or possibly learned implicitly in an end-to-end fashion [16]. Additionally, updates to these belief state-dependent action values may be encoded by dopamine neurons [11–15].

The significance of Bayesian inference becomes apparent when studying behavior in a two-armed bandit task (2ABT, Fig 1A), a serial reversal learning task where rewards for correct choices are delivered on a probabilistic schedule, and the correct port switches between two available options across blocks of trials within a single session. The fact that rewards are delivered on a probabilistic schedule generates ambiguity as to which port is currently the rewarded port following an unrewarded trial. Both mice and human subjects trained in this task are capable of maintaining stable behavior within a block, and then rapidly adapting to reward contingency changes following a block switch [17–19].

The simplest versions of standard RL models do not capture choice stability within a block and rapid switching when the rewarded port changes [18,19]. Bayesian inference may offer better models of behavior and neural computation in tasks like the 2ABT due to their ability to capture stable within-block choice and rapid between-block switching [14]. A series of studies suggested that dopamine signaling may be better explained by models accounting for belief states [9,11–13,15]. However, the additional explanatory power of Bayesian models with respect to RL models is currently ambiguous, given that a growing family of complex RL models, geared with more parameters and nuanced functions, are capable of generating adaptive behaviors [19,20]. A 2022 study of human behavior in the 2ABT found that a

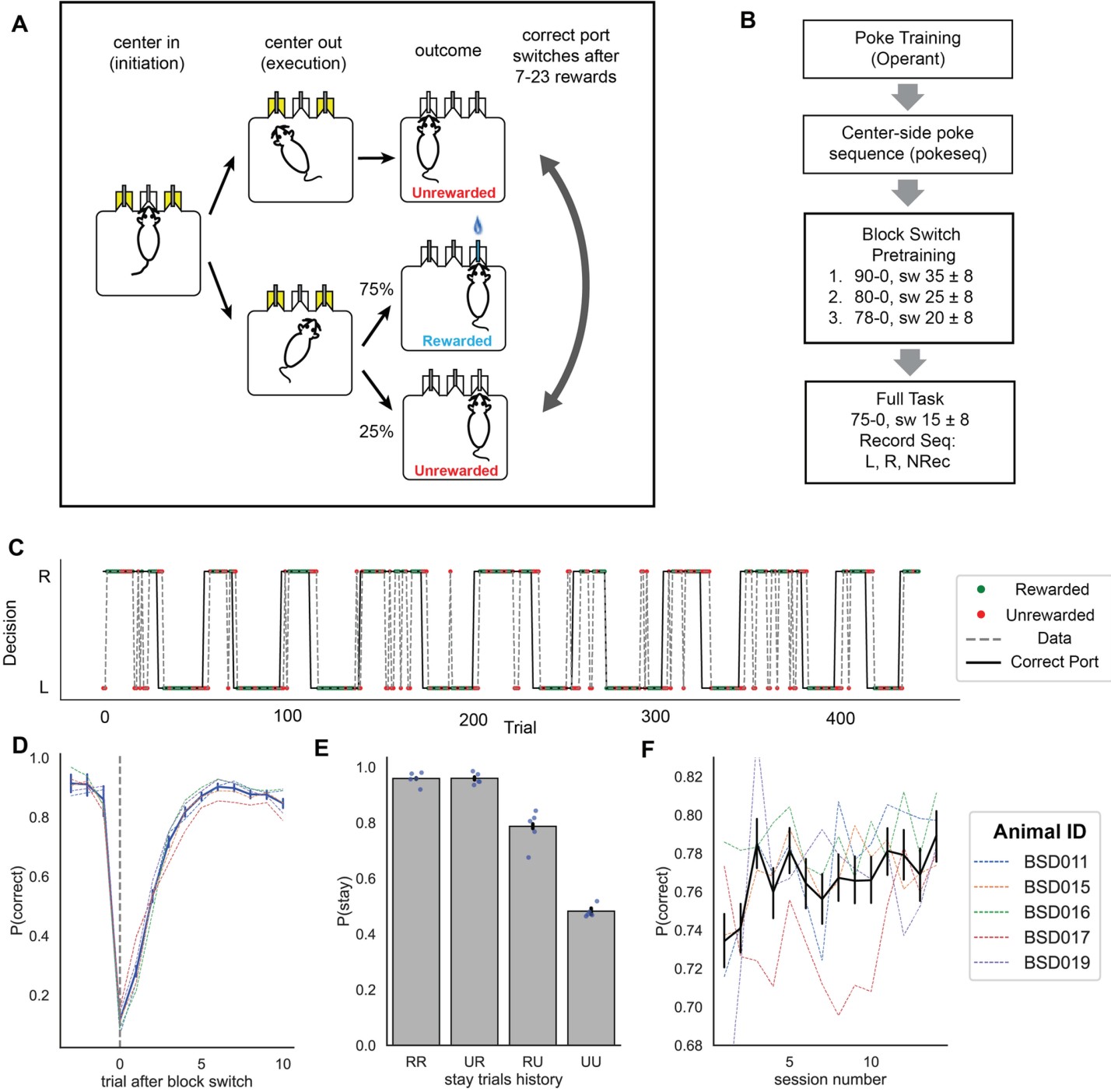

**Fig 1. Mice adapt rapidly to block switches in a probabilistic reversal task.** (A) Illustration of the two-armed bandit task, divided into initiation, execution, and outcome phases, similar to [17,31]. In the illustrated trial, the right port is rewarded with 0.75 probability and the left port is unrewarded. After 7-23 rewarded trials, the correct port switches. (B) Training protocol. The recording phase took place in the "Full Task" phase. In the pretraining phases, the structure of the task was the same as the in the full task phase, except the reward contingencies and block lengths were different. Each contingency is labeled by numbers indicating the proportion of correct and incorrect choices that were rewarded. For example, "90-0" in the first pretraining phase indicates that 90% of correct choices were rewarded. The block length in each phase is indicated by its mean and range. For example, "sw $35 \pm 8$" in the first pretraining phase indicates that switches occurred after the animal earned between 27 and 43 rewards. During the 14 sessions of mouse behavior data collection, we recorded dLight signals using a "left hemisphere (L), right hemisphere (R), no neural recording pure behavior (NRec)" sequence. (C) Raw behavioral trajectory taken from the first half of a sample session. Black line indicates correct reward port locations while dashed gray line indicates actual mouse behavior. Green dots and red dots mark rewarded and unrewarded trials, respectively. (D) Probability of making a correct choice (i.e., choosing the high probability port) as a function of the number of trials around a block switch. The vertical dashed line shows trials

at which rewarded block changes. Each colored dashed line plot shows behavioral performance for individual animals. (E) Probability of staying (repeating the last choice) after experiencing different outcome histories in the same port. RR: two consecutive rewards; UR: unrewarded outcome followed by rewarded; RU: rewarded outcome followed by unrewarded; UU: two consecutive unrewarded outcomes. (F) Performance across 14 sessions. Dashed lines show individual animal trajectories. Error bars show 95% bootstrapped confidence intervals.

Bayesian model and an RL model variant equipped with counterfactual value updates both provided complementary explanations of behavior, but neither was decisively superior [18]. A 2022 study of mouse behavior in the 2ABT suggested a purely Bayesian account of reversal learning can overestimate mouse switch probability and fail to account for choice stickiness [19].

In 2024 two further studies were published in mice that compared Bayesian and RL models ability to explain both behavioral and dLight dopamine release data from the nucleus accumbens [10,21]. Both found that behavior and imaging data were consistent with Bayesian inference. However, in both studies actions and outcomes were spread out over multiple task steps or multiple seconds, time spans that engage areas such as hippocampus and PFC [10,21,22].

There is evidence that when actions and reinforcements follow each other in more fast paced tasks, that the role of striatum is enhanced [17,23]. So, we cannot assume that computation in a fast paced 2ABT is comparable to a slower paced 2ABT even when both have latent block structure [21]. Therefore, this study provides an additional test of the robustness of Bayesian models in a faster-paced (therefore possibly more striatal based) assay of decision making.

Another limitation of the existing literature is that many studies compared only a few model variants in isolation, without including a larger growing set of models. Recently, Blanco-pozo et al. compared Bayesian models against model-free, model-based, and hybrid RL models in a two-step task [10]. Here we included a broader set of complex RL models commonly used in cognitive modeling to further elucidate how specific mechanisms, like asymmetric learning rate, counterfactual learning, dynamic learning rate, forgetting, and Bayesian inference, each differentially contribute to explaining mouse behavior and dopamine release dynamics in the nucleus accumbens in a simpler operant 2ABT task. In addition to comparing "pure" Bayesian and RL models, we can also examine hybrid models where Bayesian and RL processes are combined [11,24,25]. In these belief state RL hybrid models (BRL), Bayesian inference can be used to compute belief states, over which RL processes operate to learn policies appropriate for the current belief state.

Given these ambiguities in the past literature, our goal was to examine if Bayesian models or Bayesian-RL hybrid models could outperform sophisticated variants of RL models to explain behavior and outcome-related dopamine signals in our simple fast-paced 2ABT task with a hidden state. Mesolimbic dopamine neurons have been widely associated with RPE predictions. As one of their major projection destinations, the nucleus accumbens (NAc) is thought to play a critical role in credit assignment [26–29]. Therefore, dopamine release in NAc is of particular scientific interest. To this end, we trained mice in a value-based 2ABT task while recording dLight signals in the NAc using fiber photometry to measure dopamine transients [30]. These data allowed us to test the hypotheses regarding the cognitive process that mice employ in the 2ABT at the behavioral level and to test whether these computations are reflected by mesolimbic dopamine dynamics in the NAc.

## Results

### Mice rapidly adapt to block switches in the 2ABT

To study the flexible updating of goal-directed behaviors under probabilistic conditions, we trained mice on a 2ABT task with block switches. Two weeks prior to training, mice were injected bilaterally with AAV dLight in the NAc core and implanted with an optic fiber and ferrule above the injection site.

In the 2ABT, mice encountered three ports equipped with infrared sensors. Animals were water restricted and trained to nose-poke into the central port to initiate a trial and then move to a left or a right port to obtain a water reward. Water rewards were available on either the left or right side, depending on the block. The correct choice was rewarded with water 75% of the time whereas no rewards were available at the other port. After they received a random number of rewards (uniformly sampled from 7 to 23 for the "Full Task" condition, Fig 1A–1B), the correct port switched without any discriminative cue. In order to achieve consistent water rewards across the whole session, mice needed to readily update their choice using outcome feedback.

As soon as the full task started (see Methods for details), we recorded unilaterally from the NAc in mice performing the 2ABT for a total of 14 daily sessions, alternating the hemisphere and allowing one non-recording session every third day (Fig 1B). Just after pre-training, mice chose the high reward probability port 72.9% (95% CI: [0.669, 0.767]) of the time on average. Over the course of 14 training sessions, performance steadily increased (Fig 1F, linear regression slope coefficient 0.0026, t statistic: 2.743, p = 0.008, CI: [0.001, 0.005]). In the first 7 sessions, mice took 2.49 trials on average (95% CI: [2.39, 2.59]) to switch to the correct port after a block switch. In the 8-14th session this reduced to 2.36 trials on average (95% CI: [2.28, 2.45]). Mice switched and committed to the correct port for 3 or more trials (average 3.12 trials, 95% CI: [3.00, 3.26], S4 Fig) following the first 7 sessions. These data were comparable to mice performing the task without a fiber implant or cable [17].

To identify what task features the mice might consider before switching ports, we analyzed the stay/switch behaviors after two trials of outcome history. To simplify the comparison, we specifically chose the trials after the mice consecutively stayed in the same port for two trials or more. We encoded the outcome history at t-2, t-1 as RR, RU, UR, RU, (23607, 5719, 9428, 8520 trials each) with R representing rewarded trials and U representing unrewarded trials. For instance, a trial with "RU" stay history means that a mouse encountered a reward at trial t-2 and then was unrewarded at trial t-1, all at the same port. We found a significant reduction in the probability of repeating the previous choice after one unrewarded outcome (RR vs. RU, Fisher's exact test: 6.47, p ≤ 1e-4, Cohen's d: 0.6242, 95% CI: [0.6, 0.65]). Moreover, mice reduced their stay rate at the chosen port after two consecutive unrewarded outcomes compared to one unrewarded outcome (RU vs. UU, Fisher's exact test: 3.98, p ≤ 1e-4, Cohen's d: 0.6715, 95% CI: [0.64, 0.7]; Fig 1E).

More formally, we conducted comprehensive logistic regression analyses to describe how outcome histories influenced behaviors across individual mice, with a particular focus on capturing the decaying impact of historical reward and choice information. By implementing two distinct regression approaches (S6 Fig, [19,32]), we were able to examine how mice integrate recent (1 trial back) and more distant trial histories into their decision-making process. The first model explored the general decay of choice outcome weights across trial histories, revealing consistent patterns of how past experiences progressively influence current choices. The second model specifically evaluated whether recent rewards can "block" or reduce the effect of more distant outcome histories. This nuanced analysis allows us to parse the complex cognitive mechanisms underlying mouse behavioral decision-making, providing insight into how

mice integrate sequential information to guide their choices. By fitting these models to individual animal data, we can characterize the subtle variations in decision-making strategies across different subjects with higher precision, thus offering a more robust and mechanistic understanding of the experimental results.

## Cognitive model fitting confirms that mouse behavior data favors Bayesian models or complex RL models compared to simple RL models

To investigate the nature of computations that mice use to rapidly adapt their choices following block switches, we compared a purely Bayesian model (BIfp, see methods) adopted from [18], a hybrid BRL model (BRLfwr, see methods) inspired by [11], and several RL models (Fig 2A): a model with asymmetric learning rates for positive and negative RPEs (RL4p, dubbed the simple standard RL model), a model that additionally used counterfactual updates for unchosen option values (RLCF), the recursively formulated logistic regression (RFLR) model developed by [19], an RL model that simulates forgetting via decaying the Q value of unchosen options (RLFQ3p), a foundational dynamic learning model developed by Pearce and Hall that adapts learning rate with outcome uncertainties [33], and another recent dynamic learning variant that adjusts negative learning rate adaptively based on expected and unexpected uncertainty (RL_meta) [20]. The model space under selection consists of a rich set of cognitive mechanisms that have been shown in previous work to perform well in explaining mouse and human behaviors in the 2ABT [18,19,33], including Bayesian inference, dynamic learning, forgetting, counterfactual, stickiness, and asymmetrical learning (Table 1). We carefully chose this model space in order to identify which cognitive mechanism alone can best explain mouse behavior and dopamine release in NAc. Due to the intricate connection between these cognitive models, as well as their mathematical equivalence under certain restricted conditions [19], a definitive classification or grouping can be rather difficult. However, for the convenience and scope of this paper, we focused on evaluating the effectiveness of Bayesian inference in explaining empirical data against other more complex RL models, and opted to illustrate this with the conceptual diagram in Fig 2A. For instance, though one can show its connection to Hidden Markov Models (HMM) under restricted conditions, the RFLR model was grouped with other RL models due to its mathematical equivalence to the forgetting Q-learning model [19,34]. Modeling details can be found in the Methods and descriptions shown in Table 1.

All models listed compute choice values (or implicit values via reward probabilities in the case of BIfp), map these to choice probabilities, and then update their values (and beliefs in the case of the Bayesian models) after receiving reward feedback. For models that do not explicitly utilize RPE for model updating, like BIfp, we can calculate "pseudo-RPEs" by taking the difference between the observed and expected reward. Critically, the BRL and RL models differ in how they perform value computation. The RL models update choice values stored in a look-up table, while the BRL models computed values as a sum of choice values in each belief state weighted by posterior belief probabilities. The updating process is schematized in Fig 2D–2E.

The RL4p model and its variants have been extensively used in previous studies (e.g., [36–40]), which have provided evidence that asymmetric learning rates and choice perseveration ("stickiness") are often helpful in capturing animal behaviors. RL4p serves as a baseline against which all more complex models should be compared. Despite its past empirical success, a critical limitation of the RL4p model is that it fails to capture the observation that animals appear to update values for unchosen options (counterfactual updating; [18,41–43]).

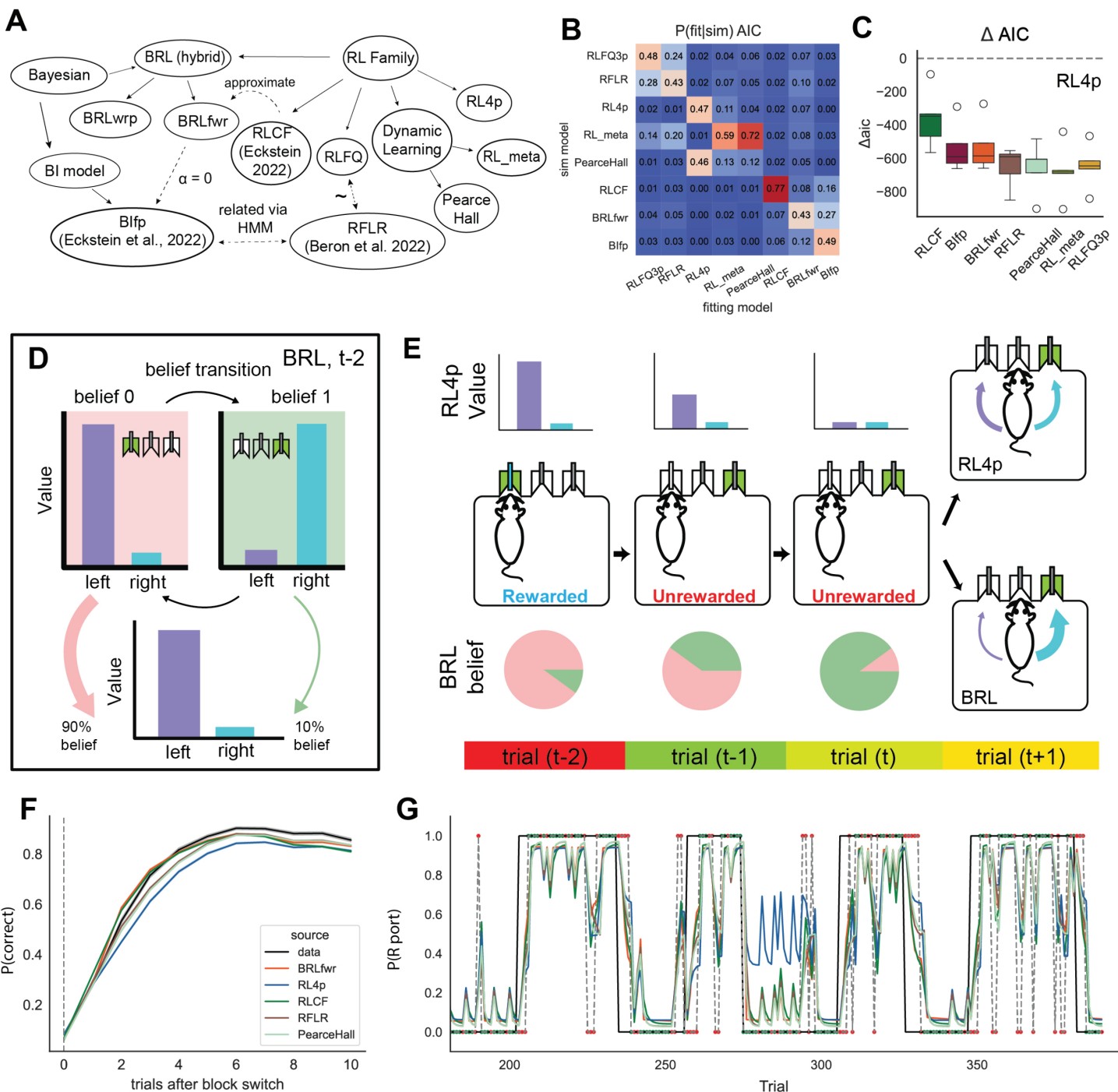

**Fig 2. Bayesian and reinforcement learning models.** (A) Relationships between cognitive models (see Methods for more details). (B) Confusion matrix outlines results for model identification analysis. Each entry i, j represents the percentage of time that the column j fitting model best explained data generated by row i simulating model. The row orders are sorted via dendrogram based on model similarity (see Methods). (C) Model comparison using relative AIC compared to RL4p: ΔAIC = AIC(model) – AIC(RL4p), with lower values indicating better fit. (D) Illustration of value computation for BRL model family, which updates beliefs via Bayes' rule and then uses these beliefs to compute values. (E) Illustration using a four-trial sequence (similar to [31]) to show the differences between RL4p and BRL. Top: purple and cyan bars show the choice values conditioned on the belief state; Bottom: pie charts show the belief state for BRL; the animal's policy is selected as a function of the value within their belief states. (F) Behavior of different models compared to mouse data (black line). Trial 0 is when the program has switched the rewarded side in a block switch. (G) Example behavioral trajectory (probability of choosing the rightward port) predicted by different models. Mouse data are marked by a dashed line and block structure is marked by a solid line. Rewarded trials are marked as green dots and unrewarded trials are marked as red dots. Error bars show 95% bootstrapped confidence intervals.

**Table 1. Overview of different cognitive models.**

| Model name | Description | Parameters | Distinct mechanisms |
|---|---|---|---|
| RL4p | Q learning | $\alpha^+, \alpha^-, \beta, \phi$ (stickiness) | asymmetric learning, perseverance |
| RLCF | Q learning with counterfactual updates [18] | $\alpha^+, \alpha^-, \beta, \phi$ | counterfactual, asymmetric learning, perseverance |
| RFLR | Recursive Formulation Logistic Regression [19] | $\alpha, \phi, \tau$ (decay rate) | perseverance, forgetting, connections to HMM [19] |
| BRLfwr | Belief state RL model with reward weight updates and fixed initial weight | $\beta, \phi, q$ (latent switch rate), $\alpha$ | perseverance, Bayesian inference |
| BIfp | Bayesian Inference Model [35] | $\beta, \phi, q$ | perseverance, Bayesian inference |
| RLFQ3p | Reinforcement Learning with Forgetting $\zeta$ (3 parameter) | $\beta, \alpha^+, \alpha^-$ ($\zeta = 1 - \frac{\alpha^+ + \alpha^-}{2}$) | asymmetric learning, forgetting |
| RL_meta | RL with meta learning [20] | $\alpha^+, \alpha^-, \beta, \phi, \zeta, \alpha_\nu, \psi$ | dynamic learning, asymmetric learning, forgetting, perseverance |
| PearceHall | Pearce-Hall dynamic learning model [33] | $\alpha_+, \alpha^-, \phi, \alpha_\nu, \zeta$ | perseverance, asymmetric learning, forgetting, perseverance |

For example, in reversal learning tasks like the 2ABT, observing an unexpected reward omission dramatically increases the likelihood of a switch, accompanied by neural responses that anticipate the new reward contingencies [9,44]. Counterfactual updating is usually formalized by updating values for unchosen actions in the opposite direction from the values of chosen actions. This qualitatively mimics the behavior of Bayesian models [35].

We fitted all models to the choice data of all 14 sessions for 5 mice (Fig 1F) in the 2ABT using maximum likelihood estimation. To qualitatively analyze similarities among this rich set of cognitive models, we simulated behaviors using parameter ranges fit to the mouse behavioral data. Then we fitted each of the 8 main model variants to each of the simulated behavioral data sets and tested for model identifiability based on the frequency with which the ground-truth model was chosen by the model selection criterion, the Akaike Information Criterion (AIC). This revealed relatively poor identifiability between the Bayesian model and the hybril BRL, as well as poor identifiability between the complex RL models (RL_meta, RLFQ3p, RFLR). These results indicate that the 2ABT cannot be used to discriminate within each family of model on the basis of behavioral data, but can discriminate across families. Furthermore, the 2ABT is adequate for rejecting the standard RL model (RL4p, Fig 2B, 2C).

With RL4p as a baseline, we found that both BRLfwr and BIfp significantly improved the AIC measure compared to the RL4p model (BIfp: $\Delta$AIC = $-536.57 \pm 59.48$, BRLfwr: $\Delta$AIC = $-530.59 \pm 61.86$). The RLCF also explained the mouse behavior better than the RL4p model (RLCF: $\Delta$AIC = $-361.92 \pm 70.52$), with a higher relative AIC on average than the BIfp model ($\Delta$AIC = $174.64 \pm 47.83$) (Fig 2C). Additionally, other complex RL models also outperformed the RL4p model ($\Delta$AIC = $-652.53 \pm 49.26$, $-677.84 \pm 65.69$, $-656.44 \pm 62.16$, $-646.45 \pm 53.62$ for RFLR, RL_meta, PearceHall, and RLFQ3p respectively, Fig 2C). When we evaluated the qualitative difference for model fitting across subjects, we noted that due to the limitation of sample size, the lowest attainable uncorrected p-value against a one-sided alternative using non-parametric pairwise tests (like permutation test) is 0.03125, which can readily lose power when multiple test corrections are used. Furthermore, it is reasonable to assume that relative AIC values across subjects follow a symmetric and normal-adjacent distribution. Therefore, we used parametric t-test and controlled family-wise error rate (FWER) via Holm's method

with Bonferroni adjustment. We found that both Bayesian models like BRLfwr (p corrected: 0.018) and BIfp (p corrected: 0.015), and complex RL models (p corrected for relative AIC for RFLR: 0.004, RLFQ3p: 0.006, for RL_meta: 0.009, PearceHall: 0.009) provided better accounts of mouse behavior in the 2ABT than the standard RL model. However, there was not sufficient discriminative power to identify whether Bayesian models or any member of the complex RL models was superior based on the simple quantitative account of all behavioral data as a whole (corrected p value for BRLfwr vs RL_meta: 0.39, BRLfwr vs PearceHall: 0.679, BRLfwr vs RFLR: 0.679, BRLfwr vs RLFQ3p: 0.679, though BRLfwr performs significantly better than RLCF: p corrected: 0.020).

## Bayesian models and complex RL models qualitatively recover mouse behaviors around trials feature volatile outcome changes

To further understand the mechanism by which the Bayesian model, hybrid BRL, or the complex RL models explained mouse behavior, we compared qualitative signatures of the BIfp, BRLfwr, and RL models with those of mouse data. One hallmark behavioral signature of mouse adaptation to block switches is improved choice accuracy as the number of trials after a block switch increases (Fig 1D). Due to the model similarities found in the previous model identification analysis, (Fig 2B, 2C), we focused on a restricted subset of model space: BRLfwr, RLCF, RL4p, RFLR and PearceHall. We omitted BIfp, RLFQ3p, and RL_meta from the analysis due to their qualitative similarity to BRLfwr, RFLR, and PearceHall respectively. Accordingly, we first compared the rate at which choice accuracy improved after block switches for simulated model behaviors against mouse data. Consistent with our hypothesis, we observed that BRLfwr and RLCF predicted faster adaptation to block switches compared to RL4p, resembling the adaptation rate of mouse switching behavior (Fig 2F). When we focused on single-trial choice-outcome trajectories, we qualitatively observed that the Bayesian model and complex RL models updated their switch probability differently (or faster in most cases) from the RL4p model after unrewarded observations or near block switches (Fig 2G). From this, one further hypothesis naturally arises: the Bayesian models and complex RL models outperform the RL4p model because of their ability to detect choice-outcome sequences that are suggestive of a block switch or volatile choice-outcome contingencies.

To test this hypothesis, it is imperative to obtain a more granular understanding of how different trial outcome histories drive a mouse to switch ports, and to determine if BRLfwr can successfully predict switch probabilities under different outcome histories, especially around block switches. Following [19], we categorized all trials based on their three past trial outcome histories, using capital A/B to denote rewarded outcomes and lower-case a/b for unrewarded outcomes. Due to the symmetrical task structure, we denoted the trial at t-3 as A/a regardless of its spatial location, and trials at t-2 or t-1 as A/a if the mouse chose the same port, or B/b if the mouse chose a different port, in reference to trial t-3 (Fig 3A). For instance, if a mouse was unrewarded at port 1 at t-3, rewarded at port 2 at t-2, unrewarded again at port 2 at t-1, we would describe the trial outcome history as aBb. We then calculated the switch probability as the probability to choose a different port at trial t compared to trial t-1.

We then compared the average mouse switch probability after each outcome history against the mean predictions of the three different cognitive models (Fig 3B; sum of squared errors (SSE): BRLfwr: 0.1796, RL4p: 0.8270, RLCF: 0.2748, RFLR: 0.1537, RL_meta: 0.0787, PearceHall: 0.1084). Notably, we observed that the RL4p model overestimated the switch probability for outcome history contexts where mice encountered rewards in both ports within the past 3 trials (e.g., ABb, AaB, etc., Fig 3A). Since these outcome contexts occurred only around block switches, mice usually stayed at the newly rewarded ports (mean switch

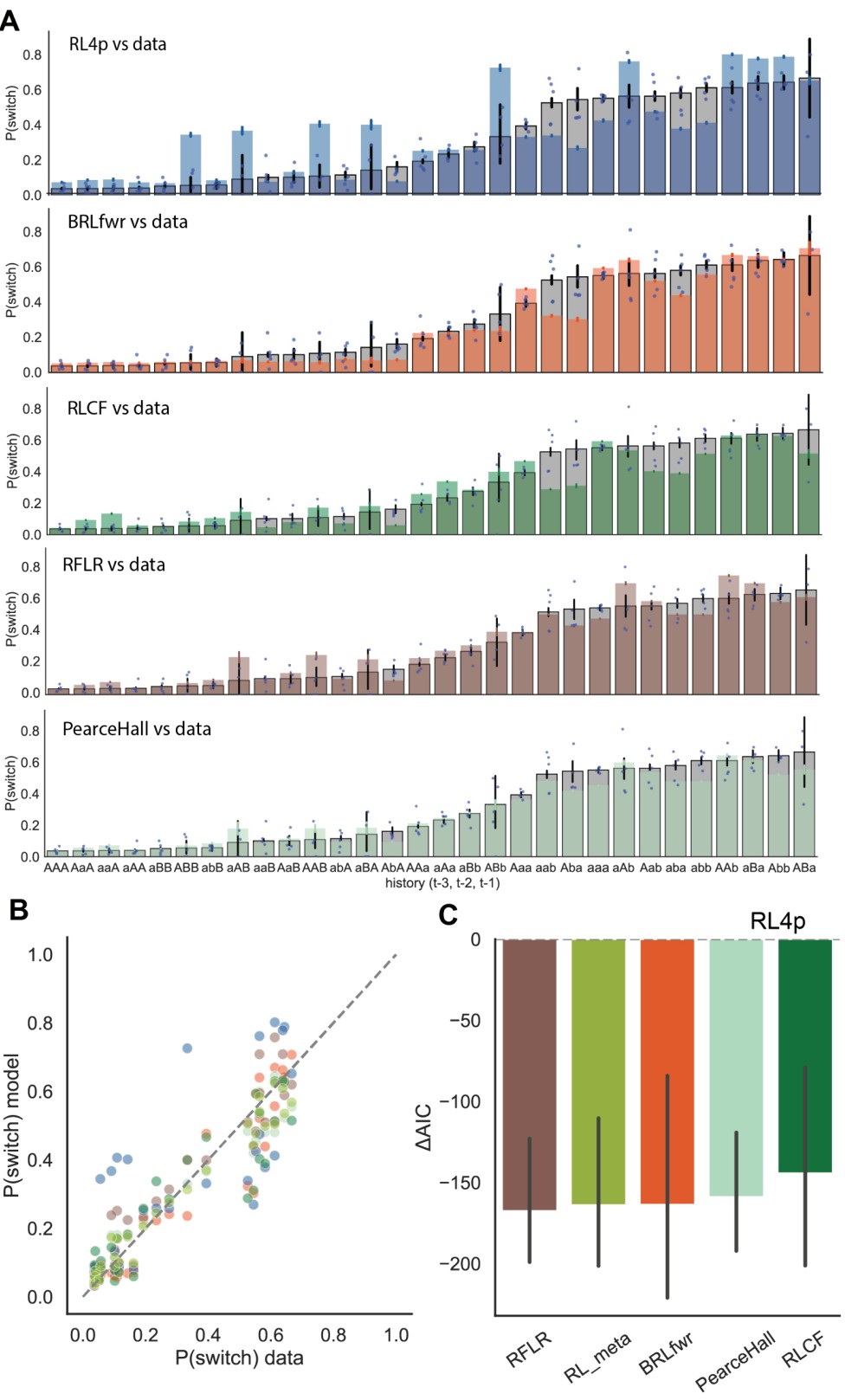

**Fig 3. BRL and complex RL models outperform standard RL by better explaining mouse behaviors around block switches.** (A) Switch probability by different trial outcome histories described by action-outcome pairs three trials back. Gray bars showed mouse average probability of switching for each outcome history, deep blue dots represent individual mice. From top to bottom: mouse data overlaid with BRLfwr, RL4p, RLCF, RFLR, PearceHall model

predictions of switch rate, respectively. (B) Switch probability predicted by different models scales with probability of mice switching port selections in different outcome contexts described. Colors represent different models, sharing the same legend as C (orange: BRLfwr, wine red: BIfp, dark green: RLCF, brown: RFLR, blue: RL4p, light cyan: PearceHall, dark yellow: RL_meta) (C) Relative AIC with respect to RL4p (dashed line at $\Delta$AIC = 0) showing model fit to mouse data around block switch. Error bars show 95% bootstrapped confidence intervals.

rate: 0.2143). BRLfwr showed a similarly low switch rate after these contexts (0.1801), whereas RL4p model predicted a high average switch rate of 0.4328 (RLCF: 0.2248, RFLR: 0.2778).

For instance, after the outcome context of ABb, the RL4p model overestimated the probability of the mice switching back to port A (Fig 3B), because it was only able to increment the action value for port B, instead of capturing the underlying reward distribution change. A mouse using a learning mechanism like RL4p would mistakenly think that selecting port A was still as valuable as it was prior to this outcome sequence, given that no reward omissions were recently experienced at that port, leading to regressive errors back to port A. Interestingly, the RL_meta model was able to explain adaptive behaviors with even higher accuracy than BRLfwr due to its adaptive power via learning rate tuning when reward context changes rapidly. We furthered our model comparison using AIC ($\Delta$AIC) relative to the RL4p standard RL model to examine mouse behaviors around block switch (defined as the first 5 trials after block switch). We found similar $\Delta$AIC results for BRLfwr and dynamic RL models (BRLfwr: −163.6763, RL_meta: −163.9207, PearceHall: −158.9066) (Fig 3C). However, RL_meta was also the most complex model with 7 parameters, with poor parameter identifiability (S2 Fig). One might therefore argue that that the Bayesian model BRLfwr was better because it exhibited good parameter recovery and was more parsimonious.

## Predictions from BRLfwr and BIfp capture nucleus accumbens core dLight dynamics better than RL models

Next, we investigated whether Bayesian models or RL models provide qualitatively and quantitatively accurate predictions of dopamine release in the NAc triggered by action outcome feedback. Importantly, as noted above, BIfp does not inherently calculate an RPE term, since it uses Bayesian inference for model updates. Therefore, to allow the comparison between model predictions of BIfp and the dopamine signals, we calculated a pseudo-RPE for BIfp, the difference between observed and expected reward.

Our experimental mice were implanted with optical fibers bilaterally in NAc core to enable recording of dLight signals. Histology images were visually inspected after the experiment to verify the implant tip location and viral expression (Fig 4A). We analyzed the dopamine responses in NAc by aligning dopamine signals to the "outcome" event, the time point when water rewards were either delivered to the mice or were omitted. After mice received a water reward, we observed a dLight dopamine signal (Z(DA)) increase on average (estimate: 1.6345, CI: [1.613, 1.656], $p \leq$ 1e-5) in ports both contralateral and ipsilateral to the recording hemisphere, consistent with previous literature [7,45,46]. When reward was omitted at the peripheral port, we observed a reduction (OLS estimate: −1.5840, CI: [−1.604, −1.564], $p \leq$ 1e-5) in the dLight signal in NAc (Fig 4B–4C).

To test the effect of animal choice switching, choice laterality (with respect to dopamine recording hemisphere), and reward or reward omissions on dopamine responses at outcome phase, we fitted a simple OLS model with all four factors. We observed a significant Z(DA)

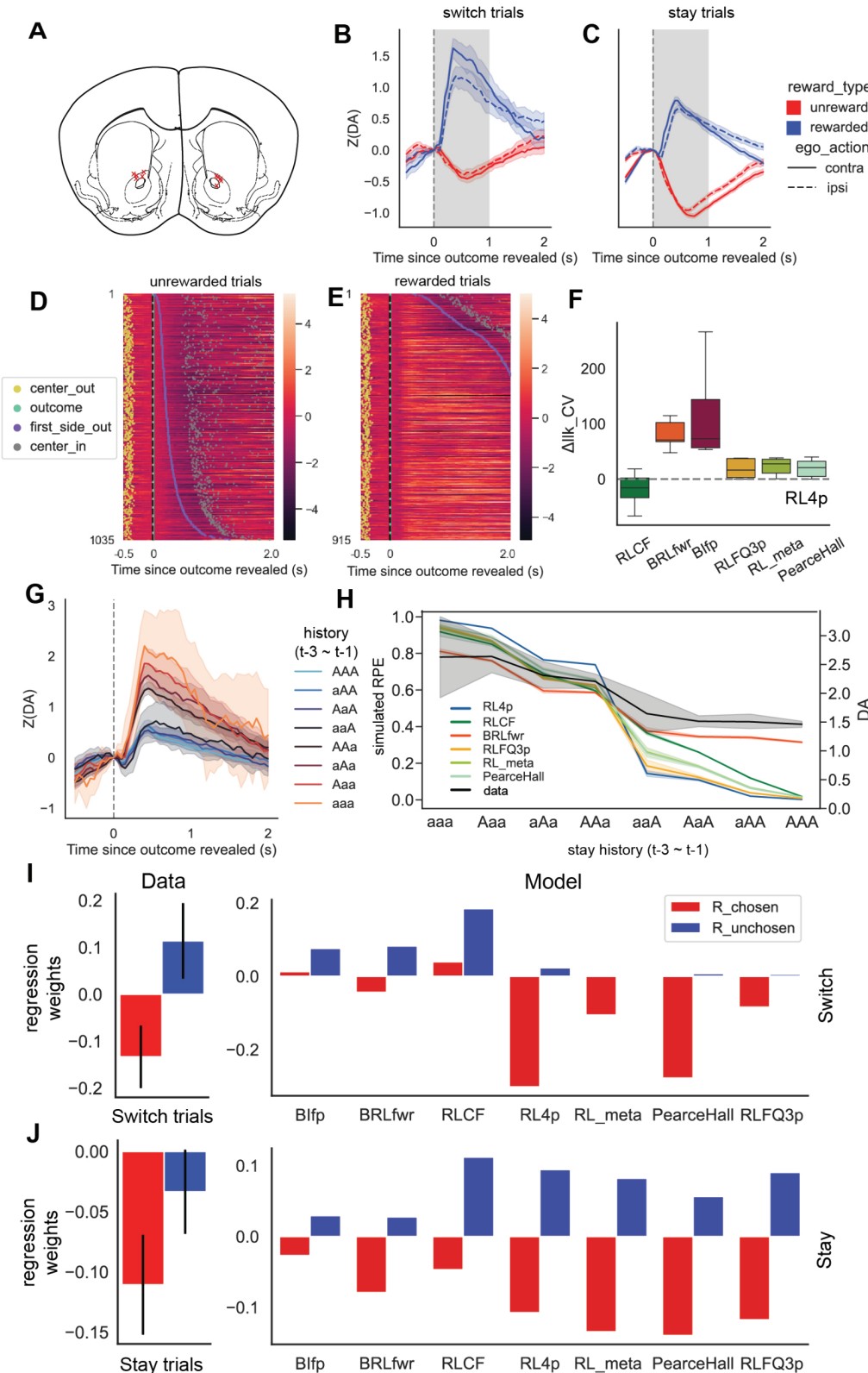

**Fig 4. NAc dLight dopamine dynamics consistent with RPE predictions by models with Bayesian inference.** (A) Implant fiber locations indicated on mouse brain atlas with red crosses, similar to [31]. (B-C) Trial average of NAc dLight signals (z-scored, as described in Methods) aligned to outcome events. Shaded area indicates the one second where the peak or trough is taken for neural regression. (B) shows switch trials and (C) shows stay trials. Rewarded

trials are in blue and unrewarded trials are in red. Trials where mice picked the port contralateral to the recording hemisphere are plotted with solid lines; trials in which mice picked the ipsilateral port are plotted with dashed lines. (D-E) Example session single trial dLight responses plotted in heatmaps, trials sorted by the time mice spent in the reward port (see Methods for further details). (D) shows a heatmap for unrewarded trials, and (E) shows rewarded trials. Increase in dLight signal is indicated by brighter shades of red and decreases from baseline are indicated by darker shades of black. Dots are used to mark "center out" (yellow), "outcome" (green), "first side out" (purple), "center in" (gray) events, respectively. (F) Result of neural regression using model RPE values to explain dopamine variability. Fit is measured as cross validated log-likelihood (llk_CV) relative to the RL4p model, with higher values indicating a better fit. Gray dashed line indicates the baseline of RL4p RPE fitted to dopamine measurements. (G) Dopamine response on rewarded trials binned by past history, sorted in increasing order of number and recency of rewards (note in all cases mice stayed with the same port 'a/A' for all three trials). (H) RPE predictions from different models plotted against dopamine peak values (in black). (I) Left: Relative change in dopamine as R_chosen (past rewards observed at the selected port) and R_unchosen (past rewards observed at the opposing port) change, calculated via LMER regression weights for dopamine observed in trials where the animals switched their port choices (animal switch trials). Right: Relative change in model RPE as R_chosen and R_unchosen change, calculated via regressions using model RPE predictions. (J) Similar to I, but for trials where the animal maintained their previous port selections (animal stay trials). Error bars show 95% bootstrapped confidence intervals.

difference (OLS coefficient estimate for switch: 0.5030, CI: [0.454, 0.552], $p \leq$ 1e-5) in trial-averaged dLight responses to the "outcome" event between switch trials and stay trials. Signals triggered by reward were larger on average on trials where mice switched to a new port, compared to when they stayed with their past choice (Fig 4B, 4C). Negative signals observed after reward omissions in trials where mice stayed were larger when compared to when they had just switched (Fig 4B, 4C). Our data were consistent with expectations that rewards are followed by increases in dopamine release in NAc (estimate: –1.6345, CI: [1.613, 1.656], $p \leq$ 1e-5) and unrewarded outcomes with a decrease (estimate: –1.5840, CI: [–1.604, –1.564], $p \leq$ 1e-5). There was no significant effect of hemisphere (laterality, contra or ipsi relative to reward port) on NAc dopamine release ($p = 0.523$) (Fig 4B, 4C).

When we visualized single trial dopamine traces around "outcome" events, we found that the temporal span of the dopamine response qualitatively aligned with the amount of time mice stayed at the peripheral reward port (Fig 4E). Similarly, after unrewarded outcomes, we found that the duration of the decrease in dLight response qualitatively aligned with the time mice stayed in the peripheral reward port, reaching signal trough typically just after mice left the port. These results together suggest that when modeling dopamine responses, the time duration that mice stayed in the peripheral reward port after an "outcome" event (dubbed "port duration") needs to be taken into consideration (Fig 4D–4E). For this reason, we included port duration as a covariate in our regression analyses.

We reasoned that if the internal reward prediction updates of the mice are represented by dopamine in the NAc and resemble that of BRLfwr or BIfp, then BRLfwr or BIfp RPE calculations should explain a larger amount of variance in the dopamine responses in NAc at the time of "outcome" events than RL models. To simplify the testing procedure, we obtained dopamine summary statistics by taking the peak (as the maximum value) of dopamine transients for rewarded trials and troughs (as the minimum value) for unrewarded trials (dubbed DA-PT) during the one second window after the outcome, marked in gray in Fig 4B. To control for behavioral confounding variables, we included session number, port duration, egocentric action, movement time, and center port poke duration as covariates in addition to predicted RPE values from BIfp, BRLfwr, RL4p, and RLCF (note that we omitted RFLR model since it does not have a straightforward RPE formulation), each in a disjoint regression covariate set. Using maximum likelihood estimation with featurized covariates (see Methods) on the five different RPE covariate sets, we found that both BIfp and BRLfwr models yielded larger relative log-likelihood (Δllk CV) than the RL4p baseline, evaluated on cross validation

sets (70-30) for each animal (Fig 4F, Δllk CV: BRLfwr: 80.58, 95% CI: [60.84, 102.79], BIfp: 118.64, 95% CI: [65.71, 217.50]). Both Bayesian models also outperformed other complex RL models on this measure (Δllk CV: RLCF: -19.23, 95% CI: [-47.03, 4.64], RLFQ3p: 18.85, 95% CI: [4.98, 33.31], RL_meta: 22.54, 95% CI: [7.96, 33.90], PearceHall: 19.50, 95% CI: [6.04, 32.83]).

To illustrate the quantitative differences between model predictions, and to identify the aspects of the data that the BRL and BIfp models captured but the RL models did not, we investigated conditions where different models gave qualitatively different predictions. First, we looked at the changes in dLight responses to rewards as mice experienced a series of outcomes in consecutive stay trials from aaa to AAA (lowercase "a" indicates unrewarded and uppercase "A" indicates rewarded). As mice encountered more rewards in their recent history, dLight dopamine responses to reward gradually decreased (consistent with decreased RPE) but did not completely flatten to no response (Fig 4G), partially explaining the quantitatively worse fit (Fig 4F). This pattern observed in our dLight data from the NAc core ('data' indicated by a black line) was not well captured by the RL models in later trials but was captured by the Bayesian model and the hybrid BRL model. To quantify, we trained a separate OLS model for each cognitive model to predict dopamine response using only simulated RPEs. We compared the model fitness across different cognitive models with cross validated log-likelihood (BRLfwr: –985.89, RL4p: –991.80, RLCF: –992.87, PearceHall: –993.81, RLFQ3p: –994.74, RL_meta: –995.02) (Fig 4H). Indeed, BRLfwr RPE predictions achieved the highest model fitness to dopamine data. The RL models updated action values on each trial as mice encountered consecutive rewards, estimating a recent average of choice outcomes. This feature may make RL models overly sensitive to recent outcome histories and liable to overestimating the action values, particularly around block switches. In our dataset, the best fitting RL4p model had an average $\alpha^+$ of 0.85 (95% CI: [0.77, 0.94]), and average $\alpha^-$ of 0.73 (95% CI: [0.71, 0.75]). The large $\alpha$ (learning rate) values likely enabled RL4p to maintain the flexibility to adapt to block switches, but they also posed issues regarding biological feasibility. RL_meta and PearceHall model were able to mitigate this issue with a dynamic learning rate that adapts to outcome uncertainty. However, due to its iterative nature, as the amount of reward in the recent outcome history increased, it steadily decreased the RPE towards zero, which did not match what we observed in the dopamine signal.

Both BIfp and BRLfwr were able to explain the adaptation of dopamine transients after increasing consecutive rewards better than the RL models. These models accounted for possible block switches and maintained values of both a high reward context and a low reward context that are not associated with any given port (Fig 4H). Instead, these values were assigned to specific ports in proportion to the model's belief state on any given trial. We speculate that this information about task structure provided robustness in response to probabilistic reward omissions after selecting the more rewarding port. This also allowed the best fitting BRLfwr and BIfp model to predict a small but non-negative RPE response on rewarded stay trials (Fig 4H), suggesting a degree of uncertainty as to whether the block had switched and capturing a pattern observed in the NAc dLight signals in later consecutive trials (right side of Fig 4H). Furthermore, we observed that among trials after an immediate past trial reward, the dopamine response was highly consistent, independent of more distant outcome histories. Only BRLfwr captured this phenomenon qualitatively, in striking contrast to other complex RL models.

The Bayesian model and the hybrid BRL model were able to modulate expectations about both choices using inference based on one single choice outcome. This led to the prediction that at the outcome phase of each choice, an increasing number of rewards gathered at the

opposing port in past trials will reduce the RPE signal reflected in dopamine, especially trials where animals switched to a different port from the previous trial (animal switch trial). To investigate if this prediction was upheld in the dopamine data, we fitted both the mouse data and the model simulated data with linear mixed effects models (LMER, see Methods). Specifically, we modeled the dopamine values, or the predicted RPEs for simulated data, as a linear weighted sum of current reward (Reward), rewards observed at the currently chosen port (trial t) over the past 4 trials (R_chosen) and past rewards observed at the opposite port over the past 4 trials (R_unchosen), conditioned on whether animal switched trials or animal stayed with their prior choice t-1 on the current trial t. On animal switch trials, both BIfp, BRLfwr, and RLCF predicted a positive effect of R_unchosen, but other RL models predicted little to no effect due to a forgetting effect or asymmetrical value updates. LMER fitting revealed a significantly negative effect of R_chosen on dopamine at outcome phase during animal switch trials (slope estimate: –0.128, 95% CI [–0.195, –0.062], p = 0.02), and a significantly positive effect of R_unchosen on dopamine during animal switch trials (slope estimate: 0.126, 95% CI [0.035, 0.217], p = 0.03, Fig 4I). On animal stay trials, the result was more complicated. All models predicted a negative relationship between R_chosen and RPE signal, matching the dopamine observation (slope estimate: –0.09, 95% CI: [–0.11, –0.07], p = 0.01). However, low sample size of high R_unchosen value trials resulted in a noisy negative estimate of effect of R_unchosen, which none of the models generated consistent predictions (Fig 4J). This elevated level estimation errors for R_unchosen made it a bad target for model arbitration, while the other three measures may be a more credible source of evidence. To sum up, the relationship between past reward history and dopamine values qualitatively matched the prediction of the BRLfwr model for switch trials and the predictions of BIfp or RLCF model for stay trials. Taken together with the functional similarity of BIfp and BRLfwr (Fig 2B), we conclude that the dopamine data matched the RPE predictions of Bayesian models quantitatively and qualitatively, but we could not further discriminate between BIfp or BRLfwr.

## Discussion

Our results showed that mice were capable of rapidly adapting to block switches in an instrumental task with probabilistic and volatile contingencies, without sacrificing the robustness of their choice policy when encountering stochastic unrewarded outcomes after correct choices. Both the pure Bayesian inference model and the belief state RL model were able to capture this balance and outperformed standard RL models in explaining mouse behavior (Fig 2C, Fig 3C). While behavioral data alone was not sufficient for us to arbitrate among Bayesian models, hybrid BRL, and complex RL models, models using Bayesian inferences (both BIfp and BRLfwr) for state estimation were able to provide a more accurate account of the dopamine release events in the NAc core following choice outcomes (Fig 4G–4J). Critically, both Bayesian and hybrid model had explicit belief state representations of different block identities (Fig 2D, 2E), which provided a mechanism for encoding the rapidly changing reward contingencies in the 2ABT task. This mechanism successfully explained the ability to efficiently switch after consecutive unrewarded outcomes while also disregarding occasional reward omissions within a block. The RL component in the BRL models allowed the model to use RPEs to update its mapping from choices to values conditioned on belief states. However, we did not find decisive evidence that BRL models explained the mouse data better than pure Bayesian models. The principal conceptual advantage of BRL models is that they make direct use of RPE signals, whereas pure Bayesian models do not. This highlights the importance of neural data in adjudicating between these models.

Building on the behavioral modeling, we showed that the Bayesian and hybrid models do a good job predicting NAc dopamine activity at the time of outcomes (reward or reward omission). In particular, RPEs generated by the BRL model and pseudo-RPEs derived from the BI models qualitatively and quantitatively match the history-dependent pattern of dopamine activity (Fig 4F–4J). This finding supports a growing literature showing that dopamine signaling of RPEs depends on a belief state (or some approximation of a belief state; see [16]) in partially observable tasks [11–15]. Furthermore, our results show that in the context of a probabilistic instrumental task, dopamine release in NAc compute RPE-like signals not just based on observable input, but also internally generated state information. This confirms an expanding literature showing that "model-free" RL computations in the midbrain and striatum take in higher-level inputs such as beliefs about latent state [9,11,21,25], as well as information about reward outcomes and reward expectations [47–50].

The BRL model is closely related to structure learning models that learn multiple context-dependent policies. Such models often assume some form of Bayesian inference at the more abstract level, with RL supporting learning of specific policies [24,51,52]. Theoretical work, supported by experimental findings, has also shown that dynamic adaptation at the more abstract level of belief states/latent contexts/rules may also be supported by RL-like computations [35,53,54]. The current work cannot dissociate these possibilities.

There are several limitations to our work that can be addressed in future investigations. Here we adopted a 75-0 reward probability design from the legacy of prior lab literature. In this design, in any given block, an inferior port always yielded no reward. Consequently, mice rarely switched to a new port and got rewarded after observing rewards in the previously chosen port. Previous work has noted that inference-based cognitive models are capable of decreasing the expectations about alternative options after observing a reward for a selected option [9,21,44]. Therefore, mice using inference would exhibit a higher RPE following a reward in a new port after also observing a reward in the old port, a reward-switch-reward sequence. Yet in our dataset, there are only 94 trials initiating such sequences out of 30,707 trials in total, across multiple animal sessions, giving us limited power to address this prediction.

In our task design, mice can initiate new trials at their discretion after the outcome, with no instructed delays. Especially after unrewarded outcomes, mice left the peripheral port shortly after, approaching the center port for a new trial. As a result, in trials when mice left the peripheral port too soon, dopamine responses coincided temporally with the dopamine ramp associated with approaching center ports [55,56]. To mitigate this, we only analyzed trials with specific port durations (see Methods), which limited our statistical power. More generally, the scaling effect of port duration on dopamine could reflect a multiplexed role of dopamine in vigor, action initiation, or other functional diversity of dopamine neurons [57–65]. Also, our work does not directly address the possibility of other alternative computational hypotheses about dopamine, like directly setting the adaptive learning rate [66] or signaling retrospective inferences about causal targets [67].

Another caveat of our experimental design was that it incorporated a multi-stage pre-training protocol before we started recording dopamine data in the "full task" condition. Mice started with blocks with 100% reward probability, which was brought down in stages to 90%, 80% and then 78%, before the "full task" at 75%. Since this pre-training took approximately 10 days on average, it may have been enough for mice to develop Bayesian inference strategies before dopamine recordings started. When we fit models to single session behavioral data during the "full task" phase, we did not observe any changes consistent with transitions from one cognitive strategy to another across days in the full task (S7 Fig). As a result, our current dataset cannot address the question of whether, in the earliest days of pretraining, mice

transitioned from initial reinforcement learning to a more sophisticated belief-state inference approach.

A final limitation of our study is the fact that the RPE that correlated with dopamine signaling only played a minimal role in predicting behavior, as evidenced by the fact that BRL models did not outperform pure Bayesian models with no RL component. We believe that better discrimination between these models can come from theory-guided experimental design, possibly using a more complex task that places a greater demand on learning policies within each belief state [68,69]. Moreover, in future work, more careful examinations of distinct cognitive mechanisms may also allow us to develop new models to further illuminate the questions raised in our study.

## Conclusion

In natural environments, animals and humans often experience ambiguous outcome feedback about the hidden reward structure of the environment. We emulated this in a two-armed bandit task with switching reward blocks, and showed that mice are capable of adapting rapidly to changes in hidden reward structures after observing specific outcome feedback sequences. Computational model fitting to behavioral data suggested that models performing Bayesian updating of beliefs better explained mouse behavioral data than standard RL models. These Bayesian models also quantitatively and qualitatively matched the dopamine release in the nucleus accumbens core better than the non-Bayesian alternatives. Together, we conclude that probabilistic belief updates are critical to behavioral adaptation and RPE signaling in the mesolimbic dopaminergic system during instrumental learning in a partially observable environment.

## Materials and methods

### Ethics statement

We conducted all animal procedures following the principles outlined by the NIH Guide for the Care and Use of Laboratory Animals. We had approval from the Institutional Animal Care and Use Committee (IACUC) at UC Berkeley and the protocol number is AUP-2015-11-8145.

### Animal protocol

Mice (all C57 Bl/6 male, bred in-house aged 104-167 days) were housed on a 12 h reversed light-dark cycle (lights on at 22:00) with nesting material. Prior to surgery they were group housed with access to food and water ad libitum. We conducted all animal procedures, which follow the principles outlined by the NIH Guide for the Care and Use of Laboratory Animals, according to the protocol (AUP-2015-11-8145-3) approved by the University of California, Berkeley Institutional Animal Care and Use Committee (IACUC) and Office of Laboratory Animal Care (OLAC).

### Surgery protocol

Mice were anesthetized with isoflurane gas for stereotaxic surgery. Meloxicam was given on the day of surgery and daily for 48h after. Coordinates for the Nucleus Accumbens Core were bregma coordinate: 1.20mm anterior, ± 1.2mm medial-lateral, -4.1mm ventral. AAV-CAG-dLight1.3b or AAV9-syn-dLight1.2b were injected using a Nanojet II (Drummond scientific). Neurophotometrics NA 0.37 or 0.48 400$\mu$m optic fibers were placed approximately 100$\mu$m above the injection site. Dental cement was used to secure the implant to the skull. During a recovery period of 7-14 days, mice were singly-housed and fed ad libitum.

## Histology and imaging

Following behavior, animals were transcardially perfused with 4% paraformaldehyde (PFA) in an 0.1M phosphate buffer (PB) solution (pH = 7.4). Brains were collected and fixed overnight, followed by a transfer to 0.1M PB. To visualize striatal photometry fibers, perfused brains were sliced coronally at $50\mu$m using a vibratome (VT1000S Leica Biosystems; Buffalo Grove, IL). Immunohistochemistry was performed to amplify the GFP signal (1:1000 chicken anti-GFP, Aves Labs, Inc.; GFP-1020 followed by 1:1000 goat anti-chicken AlexaFluor 488, Invitrogen by Thermo Fisher Scientific; A11039). Slides were mounted on slides with Fluoromount-G (Southern Biotech). Slices anterior and posterior to the fiber tract were imaged at 10X on an AxioScan Z.1 fluorescent microscope (CRL Molecular Imaging Center, UC Berkeley) to confirm targeting. Detailed resources are fully described in Table.

## Probabilistic switching 2ABT task

We trained 5 male mice on a 2ABT behavioral task in which the location of the water reward was periodically switched between the two potential rewarded ports at random intervals. Trials were initiated by a nose poke in the center port and concluded after the mouse subsequently chose one of two reward ports, located on either side of the center port. Nose pokes were detected by a infrared photodiode. Once the initiating poke was sensed, lights that encompass the reward ports were triggered to cue the animals to the viability of a potential water reward. The mouse receives a 2 $\mu l$ water reward or no reward depending on their correct or incorrect, respectively, port choice for each trial.

Only one peripheral port was rewarded at a time. The setting for the number and frequency of water rewards was altered during each phase of the task. Teaching animals the task, we started them in the "operant" phase, where water rewards were offered in the peripheral ports, and then moved to the "pokeseq" phase, where mice learned to poke the middle port before going to retrieve a peripheral port water reward. Successful learning during this phase was measured as the mouse getting more than 700 rewards in less than five hours. Some food pellets were placed into the arena while the mice were learning the task to motivate them. We omitted mice from the experiment if they didn't perform successfully on "pokeseq" after five days.

Once the mice learned to poke the center port to initiate the peripheral port water rewards, we no longer put food into the arenas and advance the mice onto the "learning switch" phase. During the three subphases of the learning switch, we teach the mice that only one port is rewarded at a frequency of 90%, 80%, and 78% for each subphase respectively. Additionally, we taught the mice that the rewarded port switches after obtaining a random number of cumulative rewards set within the ranges of 27-43, 17-33, and 12-28 for each subphase respectively. Water rewards became less predictable and more variable with each subsequent phase.

Successful learning for the three learning switch phases were set at a score higher than 69.5% on both the left and right port within four, three, and two hours for each subphase respectively. If the mice seemed unmotivated, we placed some food into the arena. Once they completed the task with food, we removed the food and had them reach criterion for their appropriate subphase.

Fibers were first introduced to the mice after they complete the last learning switch subphase and are utilized for neural recording only after the mice successfully reach the learning switch criteria once again. During the "full task", used for fiber photometry recording, mice received a water reward with a 75% chance if they picked the correct port, but received nothing if they picked the wrong port for each trial. Once mice obtained a random number of

cumulative rewards, sampled from 7 to 23 uniformly, from one port, the reward was switched to the other port. This structure minimized the possibility for the mice to predict the timing of the switch.

When participating in this task, the mice were water restricted but maintained ad libitum access to food. They were supplemented with additional water if they earned less than 500 $\mu l$ water from the task. Mice were weighed daily, before and after the task, and were given additional water if needed, at least 30 minutes after training sessions, to ensure that they maintained around 85% of their original weight.

### Fiber Photometry (FP)

We recorded from all 5 mice with fiber implants in nucleus accumbens core (NAc) using Neurophotometrics FP3001 system at 20 Hz. dLight signals are measured in 470 nm channel, while isosbestic control signal is measured in 415 nm channel for artifact removal. During the learning switch phase, we start plugging in fiber implants without recording to get mice accustomed to moving and behaving with fibers. Starting from the first "full task" session, we alternate between a left hemisphere recording, right hemisphere recording, and no recording schedule. The no recording day was implemented to avoid signal bleaching. All FP recordings are synchronized to behavioral and video data via custom TTL systems and bonsai programs and saved for further processing. Recorded signals are preprocessed using custom python program to control for motion artifacts using 415 nm reference channels [70]. Specifically, we first subtracted the channel-specific trends by approximating a smooth fit using airPLS algorithm across both channels. Then we perform a robust linear regression from reference channel to signal channel, and subtract the fitted 415 nm artifact signal from 470 nm channel recordings and obtain the processed dLight signals. All dLight signals are subsequently z-scored to account for extraneous variations between sessions.

### Task behavior analysis

All final behavioral analysis was conducted with custom implemented python packages. Behavioral data were first collected using a custom implemented 2ABT control module in matlab, and then preprocessed into behavioral data frames with each row representing a separate trial with various columns containing information regarding task meta data, mice choice data, and task relevant variables. For trials that mice failed to initiate and make a choice, we included the data but noted the relevant variables as NaNs/null. For most behavioral analysis, outcome histories as well as covariates with null entries were dropped. All model plotting are generated via `seaborn` packages, and test statistics and various measures are generated using `statsmodels`, `pingouin` and `sklearn`.

We divided the tasks into initiation phase, execution phase, outcome phase and inter trial interval (ITI) phase. The initiation phase includes both "center in" and "center out" events when mice poke their nose into and out of the center port to initiate a trial. To execute a choice, mice leave the center port and then enter one of two peripheral ports to make a selection. When the outcome of their choice is revealed to the mice, they linger at the port for water reward or time-out until they leave the port during the "side out event".

We defined the following critical events:

- Center in (CI): when mice first poke into the center port to start a new trial.
- Center out (CO): when mice leave the center port to select ports.
- Side in (SI): when mice poke peripheral port.

- Outcome (O): when outcome is revealed, typically immediately after side in (latency on the order of system latency).
- Zeroth side out (SO0): first time when mice move the nose poke out of the port trigger IR beam break.
- First side out (SO1): last time of a consecutive set of beam breaks at the same side peripheral port. Zeroth side out and first side out can often be the same when the animal leaves the port only after drinking all the water.
- Last side out (SOf): last side out beam breaks before next center port poke to initiate a new trial.

We define the following critical timing measures:

- Movement time (`MVMT`) : CI - SOf (t-1), interval between final side port departure and new trial initiation.
- Inter-trial interval (`ITI`): CI - SO1(t-1), interval between the initial side port departure and new trial initiation. This quantity is often the same as MVMT above, as mice typically do not reenter side port before a new trial after leaving side port.
- Center duration (`center_dur`): CO-CI, duration that mice lingered at center port after their trial initiation.
- Port duration (`port_dur`): SO1-O, intervals between outcome and side port departure.
- Side out bout latency (`SO_lat`): SO1-SO0, which represents the gap between two nose poke bouts at side ports. They are typically close to 0, as mice generally proceeded to the next trial after finishing drinking at the side port. Since tnose pokes are measured with infrared beam breaks, critically this is also used to control for atypical trials with excessive beam breaks.

## Computational modeling

We formalize the decision problem facing animals as follows. At time $t$, the animal makes a choice $c_t \in \{-1, 1\}$ and then observes a reward $r_t \in \{0, 1\}$. By convention, we take $c_t = -1$ to denote the left port, and $c_t = 1$ to denote the right port. The reward probability depends on the chosen action and the hidden state $z_t \in \{-1, 1\}$:

$$P(r_t = 1 | c_t, z_t) = \begin{cases} \rho_1 & \text{if } c_t = z_t \\ \rho_2 & \text{if } c_t \neq z_t \end{cases} \tag{1}$$

where $\rho_1$ is the probability of reward for a correct choice, which equals 0.75 during the full task, and $\rho_2$ is the probability of reward for an incorrect choice, which is technically 0 but which we set to 0.0001 by default to allow for model misspecification. The hidden state changes on each trial with probability $q = P(z_t \neq z_{t-1})$.

We assume a common functional form for the choice policy across models:

$$P(c_t = 1) = \sigma(\beta D_t + \phi c_{t-1}) \tag{2}$$

where $\sigma(x) = 1/(1 + e^{-x})$, $D_t = Q_t(1) - Q_t(-1)$ is the value difference, $\beta \geq 0$ is an inverse temperature parameter, $\phi \geq 0$ is a stickiness (choice perseveration) parameter, and $Q_t(c)$ is the value assigned to choice $c$ at time $t$. The stickiness component captures the tendency to repeat choices independent of the reward history.

The models make different assumptions about how the values are updated based on experience:

- **RL4p** is a standard RL model that updates Q-values according to a delta rule, with separate learning rates for positive ($\alpha^+$) and negative ($\alpha^-$) reward prediction error (RPE):

$$\Delta Q_t(c_t) = \alpha_t \delta_t \tag{3}$$

$$\alpha_t = \begin{cases} \alpha^+ & \text{if } \delta_t \geq 0 \\ \alpha^- & \text{if } \delta_t < 0 \end{cases} \tag{4}$$

where $\delta_t = r_t - Q_t(c_t)$ is the RPE.

- **RLCF** is an elaboration of the RL4p model that uses "counterfactual" updates [18]. Whereas RL4p only updates the chosen action value, RLCF additionally updates the unchosen action value in the opposite direction:

$$\Delta Q_t(-c_t) = \alpha_t \tilde{\delta}_t \tag{5}$$

where $-c_t$ is the unchosen action and $\tilde{\delta}_t = 1 - r_t - Q_t(-c_t)$ is the counterfactual RPE.

- **BRL** (belief state reinforcement learning) is an RL algorithm that computes the values as a linear function of the belief state $b_t(z) = P(z_t = z | c_{1:t-1}, r_{1:t-1})$, the posterior probability over the hidden state given the choice and reward history (the belief state update will be described further below):

$$Q_t(c_t) = w_1 b_t(c_t) + w_2 b_t(-c_t), \tag{6}$$

where $w_1$ and $w_2$ are learnable weights. Intuitively, the action value $Q_t$ for a given action $c_t$ is composed of two terms: the first term captures the reward weighted by probability when the animal is correct ($z_t = c_t$), and the second term captures the reward weighted by probability when the animal is incorrect ($z_t = -c_t$). If the animal has perfectly learned the task, the weights should be dictated by the ground truth reward probabilities, $w_1 = \rho_1$ and $w_2 = \rho_2$. We formalize a model of learning based on gradient descent, which allows the animal to approximate task rewards without knowing ground truth:

$$\Delta w_1 = \alpha \delta_t b_t(c_t) \tag{7}$$

$$\Delta w_2 = \alpha \delta_t b_t(-c_t), \tag{8}$$

where $\alpha$ is a learning rate. We considered two versions of the model. The weights were initialized to $w_1 = \rho_1$ and $w_2 = \rho_2$ (note that choosing other initial conditions had relatively little effect on model fits). Note that the weights can be interpreted as representing the Q-values of correct actions conditioned on the belief, with their update rule a confidence-weighted RL update (see e.g., [71]), hence the name BRL. We also considered a restricted version of the model (**BRLfwr**) where we fix $w_2 = \rho_2$, which we found to perform just as well as the unrestricted version (S8 Fig), so we focused on the restricted version in the main results. Belief state updating followed Bayes' rule, with

$P(z|\tilde{z})$ denoting the transition probability of the hidden state from $\tilde{z}$ to $z$:

$$b_{t+1}(z) \propto \sum_{\tilde{z} \in \{-1,1\}} P(z|\tilde{z}) b_t(\tilde{z}) P(r_t|c_t, \tilde{z}) \tag{9}$$

We assume that the belief state is initialized to $b_1(z) = 0.5$. Also we parameterize $q = P(z \neq \tilde{z}|\tilde{z})$ as the transition probability parameter of the model.

- **BIfp** (Bayesian inference with fixed parameters) makes use of the same belief state as the BRL models, but where there is no updating of the weights. Essentially, $D_t = 2b(1) - 1$. This is equivalent to $w_1 = \rho_1$ and $w_2 = \rho_2$ for BRL. Interestingly, the best fitted parameter for $\alpha$ in the unrestricted BRL version (BRLfw) above was zero for most subjects, establishing an equivalence, under this 2ABT task, between BIfp and and BRLfw.

- **RFLR** (recursively formulated logistic regression [19]) is a form of RL that is mathematically equivalent to a form of logistic regression. RFLR updates the value difference variable according to:

$$D_{t+1} = e^{-\frac{1}{\tau}} D_t + \alpha r_t c_t, \tag{10}$$

where $\tau$ is a timescale parameter governing the exponential decay rate of the values. Notably, we removed the non-negativity constraint on $\phi$ and fixed $\beta = 1$ in Eq 2 for RFLR specifically, as it is consistent with the definition in [19] and yields slightly better performance.

- **RLFQ3p** (reinforcement learning model with forgetting, totaling 3 parameters) is a popular RL-based framework to model the forgetting, or Q value decay, of the non-chosen options. The formulation is similar to standard reinforcement learning model. However, Q value of the unchosen option decays as $Q_t(-c_t) = \zeta Q_{t-1}(-c_t)$. For model simplicity, we take $\zeta = \frac{\alpha^+ + \alpha^-}{2}$. Interestingly, the forgetting parameter $\zeta$ captures stickiness effect of past chosen options, as one could show their mathematical equivalence. Consequently, the RLFQ3p model we used here has low model complexity with only 3 parameters $\beta$, $\alpha^+$, $\alpha^-$.

- **RL_meta** (reinforcement learning model with "meta learning" [20] adopts a similar general framework as RL model with forgetting ($\zeta$ is the forgetting parameter), but adapts the learning rate parameter by unexpected uncertainty $\nu$ (the rate of this adaption is controlled via $\psi$). In other words, $\alpha^-$ varies, subjected to non-negativity, as a function of how surprising recent outcomes were:

$$\alpha_t^- = \begin{cases} \alpha_{t-1}^- & \text{if } \delta_t \geq 0 \\ \psi(\nu(t) + \alpha_0^-) + (1 - \psi)(\alpha_{t-1}^-) & \text{otherwise} \end{cases} \tag{11}$$

$$\nu_t = |\delta_t| - \omega_{t-1} \tag{12}$$

$$\omega_t = \omega_{t-1} + \alpha_\nu \nu_t \tag{13}$$

Moreover, expected uncertainty variable $\omega$, initialized from 0, further mediates the update of Q values as following, on top of learning rate adaptations:

$$Q_{t+1}(c_t) = \begin{cases} Q_t(c_t) + \alpha^+ \delta_t & \text{if } \delta_t \geq 0 \\ Q_t(c_t) + \alpha_t^- \delta_t & \text{otherwise} \end{cases} \tag{14}$$

- **PearceHall** (Pearce Hall model) is one of the most established reinforcement learning model with dynamic learning mechanism [33]. It adopts a similar general framework as RL model with forgetting ($\zeta$ is the forgetting parameter), but adapts a dynamic learning component $\alpha_\nu$ with a adjustment rate $\psi$. Intuitively, $\alpha_\nu$ functions as a running estimate of magnitude of RPEs. Formally, we have:

$$\delta_t = R_t - Q_t(c_t) \tag{15}$$

$$\alpha_\nu = \alpha_\nu + \psi(|\delta_t| - \alpha_\nu) \tag{16}$$

And Q values are estimated as follows, combining both dynamic learning, as well as forgetting:

$$Q_{t+1}(c_t) = \begin{cases} Q_t(c_t) + \alpha_\nu\alpha^+\delta_t & \text{if } \delta_t \geq 0 \\ Q_t(c_t) + \alpha_\nu\alpha_t^-\delta_t & \text{otherwise} \end{cases} \tag{17}$$

$$Q_{t+1}(-c_t) = Q_0 + \zeta(Q_t(-c_t) - Q_0) \tag{18}$$

## Neural data analysis

All analysis was carried out with a custom Python module for aligning, visualization, and data modeling. After baseline correction, we visually inspected both 415 nm channel recording and 470 nm channel recording, as well as the AUC-ROC scores between the rescaled baseline values against signal channel recordings. We picked sessions that meets the following conditions: AUC-ROC higher than 0.9, longer tail in distributions in 470 nm recording compared to rescaled 415 nm recordings suggesting clear fluorescence increases, and no sharp discontinuity in both channels' recordings. We use baseline-corrected and z-scored dLight signals for in-depth analysis, called Z(DA). For neural data alignment, we interpolated Z(DA) around event times and appended them to the trial level dataframe.

Aligned neural signals are appended to the trial level data frames as additional columns. Aligned Z(DA) is baselined by subtracting out its value at "outcome" time (we denote this procedure as de-base for simplicity). For trial average dopamine visualization, we first identified a list of relevant columns as grouping variables of interest, we then dropped any row with one or more NaN entries in the relevant columns. These contain trials that either missed neural data or behavioral data, happening mostly at the beginning or the end of the session, accounting for $\sim 0.5\%$ of the trials. The trial averaged plots are then plotted using seaborn with the error bar being bootstrapped confidence intervals. When reporting summary statistics, we used bootstrapped confidence intervals to characterize mean estimates. For difference comparisons, we used suitable paired or unpaired tests and reported both the testing statistics as well as effect size, and the confidence intervals associated with it. For model RPE prediction fitting to RPE, we fitted the following regression: `outcome_DA_PT ~ rpe:rpe_pos + rpe:rpe_neg + port_dur:rpe_pos + port_dur:rpe_neg + ego_action + session_num + log__MVMT + log__center_dur`. `rpe:rpe_pos` and `rpe:rpe_neg` respectively encoded positive and negative RPE responses in dopamine. `ego_action` describes the effect of movement laterality on dLight signal. We included `port_dur, MVMT, center_dur` to control any potential movement-related components of dLight signals. To minimize the effects of intertrial events on neural signals, we selected sessions with `center_dur <= 0.8` (mice lingering less than 0.8 seconds at center port), `port_dur <= 6` (intervals shorter than 6s between outcome and side port departure), `MVMT <= 3` (intervals shorter than 3s between the final side port departure and next

trial initiation). We also filtered out trials with `SO_lat01 > 1`, where the trials had gaps longer than 6s between two nose poke bouts at side ports. These trials typically consist of trials where mice revisited the port multiple times, or simply mistriggered the infrared sensors used to measure nose pokes, biasing the task structure. After this cleaning, we were left with 20952 trials ($\sim$ 68% of the dataset) across 34 sessions with valid dopamine recordings. With the processed data, we fit the above model to dopamine data, and observed consistent results across model metrics (S3 Fig).

### Linear mixed effects modeling

Linear Mixed Effect Modeling was used to capture the effect of past rewards on the current chosen port, or the opposing port. We modeled using `rpy2` with the `lme4` package with the following formula:

```
DA ~ Reward:Switch|Subject + R_chosen:Switch|Subject +
Switch|Subject
 + R_unchosen:Switch|Subject + 1|Subject + Reward:Stay|Subject
 + R_chosen:Stay|Subject + R_unchosen:Stay|Subject
```

For model simulated RPEs, we changed the output variable to be RPE predictions instead and ran regular regressions without subject level effects due to the uniform sample size in each simulated session.

To determine the optimal lags, we did cross validation and compared multiple models with distinct feature space to arrive at R+Sw model at lag 4 (S4 Fig), which was enough to capture similar levels of variance compared to L0:R. We used 4 lags because it allows us to capture a relatively high amount of past outcome levels, with only the expense of 1% variance explained. Additionally, we fit the models to individual animals, and compared them to neural data and found varied but qualitatively consistent results across animals (S5 Fig).

### Quantification and statistical analysis

Statistical analysis was performed using Python with standard packages: scipy, statsmodels, pingouin. We included n = 5 subjects with 14 sessions each. Statistical details of each analysis can be found in each figure legend, result section or correspondent method section. Error bars are 95% bootstrapped confidence intervals, and Holm-Bonferroni correction was applied when appropriate.

**Model fitting and comparison:** We fit all models to 14 sessions of all 5 mice during the Probswitch (Full task) phase using custom implemented `cogmodels` python module. The module finds the best-fitting model parameters for each animal by maximum likelihood estimation. The model initializes the parameters by randomly sampling from distributions as follows:

- RL4p: $\alpha^+ \in \text{Unif}(0,1)$, $\alpha^- \sim \text{Unif}(0,1)$, $\phi \sim \Gamma(2,0.2)$, $\beta \sim \text{Exp}(1)$
- RLCF: Same as RL4p
- RFLR: $\alpha \sim \text{Exp}(1)$, $\phi \sim \mathcal{N}(0,1)$, $\tau \sim \text{Exp}(1)$
- RLFQ3p: $\alpha^+ \in \text{Unif}(0,1)$, $\alpha^- \sim \text{Unif}(0,1)$, $\beta \sim \text{Exp}(1)$
- RL_meta: Same as RL4p, plus $\zeta \sim \text{Unif}(0,1)$, $\alpha_\nu \sim \text{Unif}(0,1)$, $\psi \sim \text{Unif}(0,1)$
- BIfp: $\beta \sim \text{Exp}(1)$, $\phi \sim \Gamma(2,0.2)$, $q \sim \text{Unif}(0,0.05)$
- BRLfwr: Same parameters as BIfp, plus $\alpha \sim \text{Unif}(0,1)$

where $\Gamma(\alpha, \theta)$ denotes the gamma distribution with shape $\alpha$ and scale $\beta$; $\mathcal{N}(\mu, \nu^2)$ denotes the Gaussian distribution with mean $\mu$ and variance $\nu^2$.

After obtaining maximum likelihood estimates, we constructed an empirical range of parameters and performed model recovery tests. We simulated behavioral data from randomly sampled model parameters for each model from the empirical ranges, and refit to the simulated data to obtain the model-fitted values for 1000 random iterations. All three model classes are able to recover model parameters with high correlation (S2 Fig). We performed model comparison by computing the Akaike Information Criterion (AIC) for each model $\mathcal{M}_i$:

$$\text{AIC} = 2k - 2 \sum_t \left( \ln P(c_t | \hat{\theta}, \mathcal{M}_i) \right), \tag{19}$$

where $P(c_t | \hat{\theta}, \mathcal{M}_i)$ is the likelihood of the data $c_t$ conditional on the maximum likelihood estimate ($\hat{\theta}$). We report AIC scores relative to RL4p, the baseline RL model. More negative scores indicate that a model explains the data better compared to RL4p.

## Model identification

The model identification tests go through three stages: model behavior simulation, model cross-fitting, and confusion matrix construction. We denote our total model set as $\mathcal{M}$.

1. **Model behavior simulation**: Similar to the previous section, sampling randomly from the empirical range of parameters constructed from the mice data fitting, we simulated behavioral data $D_k(m_i)$ for 1000 times for each model $m_i \in \mathcal{M}$, where $k$ indexes simulation runs.
2. **Model cross-fitting**: For each simulated data set $D_k(m_i)$, we fit each $m_j \in \mathcal{M}$, and get a best fitting model $m_{ki}^* | D_k(m_i)$ based on lowest AIC measure.
3. **Confusion matrix**: For all 1000 sessions, we can compute an empirical probability that the model $m_j$ best fits to $D(m_i)$, or $P(m_j|m_i)$. Then we construct a confusion matrix from $P(m_j|m_i)$, $\forall m_i, m_j \in \mathcal{M}$, where each element at row $i$, column $j$ represent $P(m_j|m_i)$.

## Key resources table

**Table 2. Summary of key resources.**

| REAGENT or RESOURCE | SOURCE | IDENTIFIER |
|---|---|---|
| **Antibodies** | | |
| 1:1000 chicken anti-GFP | Aves Labs, Inc. | GFP-1020 |
| 1:1000 goat anti-chicken AlexaFluor 488 | Invitrogen by Thermo Fisher Scientific | A11039 |
| **Bacterial and Virus Strains** | | |
| AAV5-hSyn-dLight1.2 | Patriarchi et al., 2018 [30]; Courtesy of Prof. Lin Tian (UC Davis) | Addgene Cat: 111068-AAV5; RRID: Addgene_111068 |
| AAV9-CAG-dLight1.3b | Patriarchi et al., 2018 [30], Addgene | Addgene Cat: 125560-AAV9; RRID: Addgene_125560 |
| **Chemicals, Peptides, and Recombinant Proteins** | | |
| 4% paraformaldehyde (PFA) | This paper | N/A |
| 0.1M phosphate buffer (PB) | This paper | PB |
| Fluoromount-G | Southern Biotech | Cat: 0100-01 |
| **Experimental Models: Organisms/Strains** | | |
| C57 Bl/6 male | bred in-house | N/A |
| **Software and Algorithms** | | |
| Standard python open source packages (numpy, statsmodels, pingouin, seaborn, plotly, scipy, etc.) | python | pip OR anaconda |
| photometry baseline filtering procedure | Martianova et al., 2019 [70] | DOI:10.3791/60278-v |

## Supporting information

**S1 Fig. Mice switch and commit to correct ports faster across multiple sessions.** Top row y-axis shows the number of trials mice take to switch to the correct port in a new block (trial2sw). Bottom row y-axis shows a metric of the first trial from which the animal chose the correct port and then persisted with the choice selection for 2 subsequent trials consequently (trial2asymp). First column x-axis shows the relationship between the measures and the number of blocks within a training session, demonstrating within session learning. A downward trend for both measures suggests learning and faster switches across different reward blocks within session. Second column x-axis shows that the switch measures decreased as the number of training sessions increased, describing learning across multiple sessions. This is consistent with results in Fig 1F, where accuracy improved over multiple sessions. Third column x-axis shows the same thing as the second column, but separated by animals. Error bars show 95% bootstrapped confidence intervals.
(PNG)

**S2 Fig. Parameter recovery of cognitive models.** We took the maxima and minima for each of the best fitted parameter values for all subjects, and constructed a uniform distribution for each. We then sampled from these empirical distributions for simulating behaviors and fitted best fitted parameters for each set of simulated behaviors for each model. The results are generated after 500 runs of random parameter samples for each model, shown as a scatter plot with truth parameter against fitted parameters. Each row is a different model noted at the top left corner. Outliers, defined as 6 standard deviations from mean true parameter values, were thrown out for visualization purposes but percentages were noted.
(PNG)

**S3 Fig. Comparison of fitness of model predicted RPE to dopamine data.** Similar to Fig 4F, we included results of model fitness using different metrics: relative AIC, relative BIC, relative cross validated R2 (R2CV), and relative cross validated log likelihood (llk_CV). All metrics converged on favoring Bayesian model predictions of dopamine at outcome phase. Error bars show 95% bootstrapped confidence intervals.
(PNG)

**S4 Fig. Selection of LMER models and lags.** We compared the cross validated R2 score using different feature sets, while keeping the use of dopamine as output variable. L0:R: Standard formulation using N trial back choice and reward interactions. 3R: just R_chosen, R_unchosen, Reward features. 3R+Sw: in addition to 3R features, we included interactions of whether a trial is an animal switch trial, or animal stay trials. L0:DA: we used the interactions between past trial dopamine values and choice selections. Together we found that 3R+Sw was enough to capture similar levels of variance compared to L0:R. We used 4 lags because it allows us to capture a relatively high amount of past outcome levels, with only the expense of 1% variance explained. Error bars show 95% bootstrapped confidence intervals.
(PNG)

**S5 Fig. Dopamine data are qualitatively consistent across animals.** (A) Average differences in dopamine responses to rewards or unrewarded outcomes across switch stay trials are fairly consistent across animals. (B-C) Similar to LMER regression of influence of past rewards on dopamine responses, we did OLS for each animal separately. All but one animal showed a strong qualitative resemblance to Bayesian model predictions for Switch trials. As discussed in the main text, R_unchosen effect was highly variable across animals, and had a near-zero

net effect, corresponding more to the predictions of BIfp simulation. Error bars show 95% bootstrapped confidence intervals.
(PNG)

**S6 Fig. Behavior analysis with logistic regressions and model simulations.** (A) Following the formulation from [19], we fitted logistic regression models to individual mice behavior data and identified consistent patterns of decaying choice outcome weights for more distant outcome histories. Error bars show 95% bootstrapped confidence intervals. Specifically, we used this formula ($\tilde{C}_{t-i} = 2C_{t-i} - 1$): $C_t = \sum_i \beta_i^{CR} \tilde{C}_{t-i} R_{t-i} + \beta_{t-i}^{R} R_{t-i} + \beta_{t-i}^{\tilde{C}} \tilde{C}_{t-i}$. (B) To identify the blocking effect of the reward at trial t-1 found in [32], we fitted the following regression model, where $\beta_i^{CR+}$ represent the R1-blocked coefficient:

$$C_t = \sum_{i=2}^{p} \beta_i^{CR+} \tilde{C}_{t-i} R_{t-i} \mathbf{1}[R_{t-1} = 0] + \beta_i^{CR-} \tilde{C}_{t-i} R_{t-i} \mathbf{1}[R_{t-1} = 1] + \sum_{i=1}^{p} \beta_{t-i}^{R} R_{t-i} + \beta_{t-i}^{C} \tilde{C}_{t-i}$$

(PNG)

**S7 Fig. Model fitness results for behavioral data across sessions**: To test if there was meta-learning across sessions, we fitted cognitive models to mouse behavior across multiple sessions. We did not find evidence of any significant and consistent increase in model fitness across multiple sessions. We speculate that mice were able to learn the inference structure of the task during a brief pre-training phase, when the reward probability of the high value option changed from 90%, 80%, to 78% (Fig 1B). Model fitness of Bayesian models did show some decline in some later sessions but then recovered, suggesting that there was no sustained change in behavioral strategy after extended training. Error bars show 95% bootstrapped confidence intervals.
(PNG)

**S8 Fig. BRL model with fitted weights did not have significant model fitness differences compared to BRLfwr**: To further justify that the success of Bayesian models (e.g., BRLfwr) is not solely due to setting the initial reward probability of high value state as ground truth, we fitted BRLwrp, where both rewarded weights as well as reward probabilities are fitted to data. No statistical differences were observed between BRLfwr and BRLwrp in their ability to explain mice behavior and dopamine data. Error bars show 95% bootstrapped confidence intervals.
(PNG)

## Acknowledgments

We are grateful to Scott Linderman, Bernardo Sabatini, Celia Beron, Jing Jing Li, Thomas W. Elston, Yilan Liang, Hongli Wang, Gaia Molinaro, Jaclyn Essig, and Eric J. Hu for discussion and to Chris Machle, Lexi Z. Zhou and Anthony T. Dunn for discussion and assistance with data collection. We thank Prof. L. Tian for the provision of dLight1.2 AAV.

## Author contributions

**Conceptualization:** Albert J. Qü, Christopher D. Hall, Karyna Mishchanchuk, Andrew F. MacAskill, Anne G. E. Collins, Samuel J. Gershman, Linda Wilbrecht.

**Data curation:** Albert J. Qü, Lung-Hao Tai, Emilie M. Tu, Juliana B. Chase.

**Formal analysis:** Albert J. Qü, Samuel J. Gershman.

**Funding acquisition:** Samuel J. Gershman, Linda Wilbrecht.

**Investigation:** Albert J. Qü, Lung-Hao Tai, Christopher D. Hall, Emilie M. Tu.

**Methodology:** Albert J. Qü, Lung-Hao Tai, Christopher D. Hall, Maria K. Eckstein, Wan Chen Lin, Anne G. E. Collins, Samuel J. Gershman, Linda Wilbrecht.

**Project administration:** Albert J. Qü, Christopher D. Hall, Samuel J. Gershman, Linda Wilbrecht.

**Resources:** Lung-Hao Tai, Linda Wilbrecht.

**Software:** Albert J. Qü, Samuel J. Gershman.

**Supervision:** Samuel J. Gershman.

**Validation:** Lung-Hao Tai, Christopher D. Hall, Karyna Mishchanchuk, Andrew F. MacAskill.

**Visualization:** Albert J. Qü, Christopher D. Hall, Maria K. Eckstein, Karyna Mishchanchuk, Wan Chen Lin, Andrew F. MacAskill, Anne G. E. Collins, Samuel J. Gershman, Linda Wilbrecht.

**Writing – original draft:** Albert J. Qü, Christopher D. Hall, Emilie M. Tu, Samuel J. Gershman, Linda Wilbrecht.

**Writing – review & editing:** Albert J. Qü, Lung-Hao Tai, Christopher D. Hall, Maria K. Eckstein, Karyna Mishchanchuk, Wan Chen Lin, Andrew F. MacAskill, Anne G. E. Collins, Samuel J. Gershman, Linda Wilbrecht.

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
