## [Decision Letter · Decision Letter 0]

24 Oct 2024

PCOMPBIOL-D-24-01557Nucleus accumbens dopamine release reflects Bayesian inference during instrumental learningPLOS Computational Biology Dear Dr. Qü, Thank you for submitting your manuscript to PLOS Computational Biology. After careful consideration, we feel that it has merit but does not fully meet PLOS Computational Biology's publication criteria as it currently stands. Therefore, we invite you to submit a revised version of the manuscript that addresses the points raised during the review process. Please submit your revised manuscript within 60 days Dec 24 2024 11:59PM. If you will need more time than this to complete your revisions, please reply to this message or contact the journal office at ploscompbiol@plos.org. Please include the following items when submitting your revised manuscript: * A rebuttal letter that responds to each point raised by the editor and reviewer(s). You should upload this letter as a separate file labeled 'Response to Reviewers'. This file does not need to include responses to formatting updates and technical items listed in the 'Journal Requirements' section below.* A marked-up copy of your manuscript that highlights changes made to the original version. You should upload this as a separate file labeled 'Revised Manuscript with Track Changes'.* An unmarked version of your revised paper without tracked changes. You should upload this as a separate file labeled 'Manuscript'. If you would like to make changes to your financial disclosure, competing interests statement, or data availability statement, please make these updates within the submission form at the time of resubmission. Guidelines for resubmitting your figure files are available below the reviewer comments at the end of this letter. We look forward to receiving your revised manuscript. Kind regards, Tianming YangAcademic EditorPLOS Computational Biology Daniele MarinazzoSection EditorPLOS Computational Biology Feilim Mac GabhannEditor-in-ChiefPLOS Computational Biology Jason PapinEditor-in-ChiefPLOS Computational Biology  **Journal Requirements:** **Additional Editor Comments (if provided):****Reviewers' comments:** Reviewer's Responses to Questions

**Comments to the Authors:**

Reviewer #1: The authors investigate mice's behavioral strategies during a reversal task, specifically examining whether the rats use reinforcement learning (updating the value of each state) or Bayesian inference (inferring the hidden state). To determine this, they fit observed rat behavior with multiple models and compare the goodness of fitting. In addition, dopamine signals from the NAcc were recorded as a neural measurement of RPE, and then compared to the (pseudo-) RPEs generated by the models. The results indicate that the BLfp and BRLfwr models outperform other models, suggesting that Bayesian inference plays a crucial role in explaining rat behavior. The paper is generally well-written and easy to follow, but there is a concern that the assumptions underlying the reinforcement learning models do not capture the task's structure, which makes them appear as straw man, diminishing the significance of this study.

In the task, the reward probabilities are 75% and 0% for two options, respectively. At least two critical aspects of the task structure are implicitly incorporated into the Bayesian models but not represented in most RL models. The first one is that the reward probabilities for the two options are anti-correlated. A higher reward probability on one side directly implies a lower reward probability on the other side. While the RLCF model takes this into account, most other RL models do not. Second, receiving a reward on a single trial from one side implies that this side is rewarded while the other side is non-rewarded. The Bayesian models also have access to the exact reward probabilities, which provides further precision in their predictions.

Based on these reasons, it appears that the better fit of the Bayesian models in this study primarily suggests that the rats learn the underlying task structure rather than using a Bayesian approach to estimate the hidden state or perform pure Bayesian inference.

To enable a fairer comparison, it would be valuable to include RL models that incorporate the critical aspects of the task structure, such as the BRL models mentioned in the manuscript, but estimate the hidden states in a non-Bayesian manner. Such models would help determine whether the observed behavior is better captured by Bayesian inference or by structural learning.

Other concerns:

1. It would be interesting to model the initial learning process. Rats may form a representation of the left-rewarded and right-rewarded states after repeated reversals, but likely not before the first reversal. Additionally, the exact reward probability is unknown to animals when they are making inferences, and it should be something they learn over time. If the authors could demonstrate the transition process from a RL approach to an RL-Bayesian inference hybrid or a pure Bayesian inference model, and illustrate how dopamine activity changes during this transition, it would greatly enhance the significance of this study.

2. The authors include a meta-learning model to modulate the learning rate for negative RPE. However, why not modulate the learning rate for positive RPE as well? Additionally, models such as Mackintosh's and Pearce-Hall have been successful in explaining differences in learning rates across various tasks. Including these models could provide valuable insights and strengthen the conclusions of the study. Or at least explain why not include them.

3. Line 284: how is peak of dopamine transients calculated? Please also briefly mentioned in the main text.

Reviewer #2: The goal of this research is to determine whether dopamine concentrations in the Nucleus Accumbens core reflect reward prediction errors conditioned on the belief of the hidden state of the external world. The authors proceed to answer this question by training mice in a two-armed bandit task, modeling animals’ behavior using different types of models, and performing fiber photometry recordings using a dopamine sensor in the Nucleus Accumbens core. While the results point towards the conclusion that inference-based methods explain both behavior and dopamine concentrations, as shown by previous literature, the evidence in this manuscript is a bit weak, and this research alone does not provide a conclusive answer.

The topic of whether mice can perform Bayesian inference (or an approximation of it) and if dopaminergic RPEs are conditioned on it, as opposed to the traditional model-free computations, is of interest to the community, as shown by the growing literature pointing towards that direction. The authors state that the limitations of previous research on the topic are the lack of accounting for Bayesian inference models in 2ABT and comparing only a few models. However, the authors fail to acknowledge Blanco-Pozo et al. 2024 - https://doi.org/10.1038/s41593-023-01542-x, where they tested different RL variants and also Bayesian inference models, performed fiber photometry recordings on the dopamine system, and concluded that both mice behavior and dopaminergic RPEs are consistent with a Bayesian inference account. While Qü et al. description of different RL and Bayesian inference models, how similar and different they are, can be of importance for the community, the evidence provided in this manuscript is not completely convincing. Despite authors using a wide range of models, they are not able to differentiate between models. As stated in line 144 - behaviorally, the 2ABT used in this research cannot discriminate between models, just that any other model is better than the traditional model-free RL. While using the dopamine signals to arbitrate between models points towards inference models to better explain the signals, not all analysis supports a particular model. So it is not clear what is the advantage of using all these models if one cannot arbitrate between them. The low N in this research and, consequently, the low statistical power, also limits the results.

To strengthen this manuscript, I would suggest the authors consider adding more visual descriptions of the similarities and differences between the different models. First, by providing a table with the parameter fits for each model per animal. This way, it would be easier to understand what is driving the behavioral similarities between the models and how it ranges between animals. Secondly, while the authors show some simulated behavior (Fig. 2F, G, 3A), not all the models are included, and more analysis would be useful. For example, have the analysis of Fig 1E for all the models, or have a lagged regression of stay/switch based on the past history of rewards (like 3A but as a regression, it might be easier to see more clear differences between models).

The authors classified all their models into two big families: BRL and RL family (Fig 2A). First, as a minor comment, I would suggest the authors consider a better nomenclature for all their models. With the current acronyms that are similar to each other, it makes it harder to read the manuscript and remember what is being described. Then, I am not sure if I completely agree with their grouping. Although RLCF (the counterfactual RL) does not have a Bayesian inference component, it has been previously described as an approximation of a full Bayesian inference model (Hampton et al. 2007 - https://doi.org/10.1016/j.neuron.2007.07.022). Plus, the counterfactual RL is different from all the other models in the author’s RL family, as it includes some kind of model of the structure of the task.

Regarding the regressions to predict dopamine response, my understanding from the text is that just the simulated RPEs from the models are included. However, the dopamine response that is being analyzed is also at the same time the reward is being delivered. Therefore, it is necessary to remove the effect of outcome alone (independent of prediction) from the analysis. Can the authors please show that the same results stand if current outcome is also included as a regressor?

It is not clear how R_chosen and R_unchosen are defined. The text mentions ‘‘past rewards observed at the selected/opposing port’, however, it is not described how many trials in the past, whether there is some exponential decay, etc. Have the authors considered, instead of using the history of past rewards on each port, which has the limitation of the low sample size of R_unchosen as the authors mention, looking at the direction of the RPE based on stay or switch and the outcome on the previous trial (like in Blanco-Pozo et al. 2024 - https://doi.org/10.1038/s41593-023-01542-x), also adding the effect of the current reward as a separate effect as explained before.

In the methods section, it is mentioned that some trials were dropped. Please can the authors add what is the proportion of these? For the photometry analysis, it seems that not all the data was included, can the authors also specify how many sessions were excluded? And from the included, how many trials were dropped?

Have the authors considered adding the asymmetric impact of rewards and unrewards into their Bayesian models? For example, in their BRLfwr, have they tried to include asymmetric learning rates?

Can the authors please explain why they baseline-corrected their z-scored dLight signals? Also, why subtract the signal at outcome time, which is the time when a lot of information about the state of the task is being provided?

Minor comments:

line 307: I would not say VTA, as there is a full debate into the relation between VTA dopamine activity and axonal activity and concentration in the striatum. Maybe change for NAc core dopamine concentrations.

line 692: ‘Probswitch phase’ —> I guess that refers to the full final task?

**Have the authors made all data and (if applicable) computational code underlying the findings in their manuscript fully available?**

Reviewer #1: **No: **The code is currently not publicly available.

Reviewer #2: Yes

PLOS authors have the option to publish the peer review history of their article (what does this mean?). If published, this will include your full peer review and any attached files.

Reviewer #1: No

Reviewer #2: No

 **Figure resubmission:**While revising your submission, please upload your figure files to the Preflight Analysis and Conversion Engine (PACE) digital diagnostic tool, https://pacev2.apexcovantage.com/. PACE helps ensure that figures meet PLOS requirements. To use PACE, you must first register as a user. Registration is free. Then, login and navigate to the UPLOAD tab, where you will find detailed instructions on how to use the tool. If you encounter any issues or have any questions when using PACE, please email PLOS at figures@plos.org. Please note that Supporting Information files do not need this step. If there are other versions of figure files still present in your submission file inventory at resubmission, please replace them with the PACE-processed versions. 
---

## [Decision Letter · Decision Letter 1]

26 Mar 2025

PCOMPBIOL-D-24-01557R1

Nucleus accumbens dopamine release reflects Bayesian inference during instrumental learning

PLOS Computational Biology

Dear Dr. Qü,

Thank you for submitting your manuscript to PLOS Computational Biology. After careful consideration, we feel that it has merit but does not fully meet PLOS Computational Biology's publication criteria as it currently stands. Therefore, we invite you to submit a revised version of the manuscript that addresses the points raised during the review process.

Please submit your revised manuscript within 30 days May 26 2025 11:59PM. If you will need more time than this to complete your revisions, please reply to this message or contact the journal office at ploscompbiol@plos.org. Please include the following items when submitting your revised manuscript:

We look forward to receiving your revised manuscript.

Kind regards,

Tianming Yang

Academic Editor

PLOS Computational Biology

Daniele Marinazzo

Section Editor

PLOS Computational Biology

**Reviewers' comments:**

Reviewer's Responses to Questions

Reviewer #1: I appreciate the efforts the authors have put in, and they addressed most of my concerns. The paper should be ready for publication once the following question is addressed:

Could you clarify if Fig. S7 is fitting dopamine responses or behavior? I assume it is dopamine, but this should be explicitly stated in both the figure legend and the manuscript. While I understand that the learning process is not the focus of this study, it would still be valuable to fit behavioral data from the pretraining period, even though dopamine data were not collected during that time.

Reviewer #2: This is a revised version of a previously submitted manuscript, in which the authors analyze a wide range of models, Bayesian and more complex RL models, against the simplest RL model to explain mice behavior and dopamine levels in the Nucleus Accumbens core in a 2ABT. This revised version’s main changes are in the introduction, which places this work within the context of previous literature and clearly states the aims of the work. The authors have also added another model and extra analysis in supplementary material, which I believe helps in the comparison between models and data.

I would like to thank the authors for their replies and for taking into account my comments. I agree with the authors that the wide range of models and replication is useful and important for the community and is within the scope of PLOS Computational biology. Although their results are consistent with previous literature, showing that Bayesian models can explain behavior and dopamine signals, my main critique was that this work, despite using a wide range of models, is not able to conclusively distinguish between Bayesian and more complex RL, even if Bayesian models seems to better (but not completely) explain dopaminergic RPEs than the other RL models. I think it further highlights the complexity of modeling mice behavior and dopamine signaling using these models, and that probably, a combination of these models (or even new models) might be necessary to fully capture behavior and dopamine.

That’s why I think the claims of the manuscript, especially at the beginning of the discussion, do not precisely align with their results, and I think it could be further revised to highlight the complexity and difficulty in fully capturing behavior and dopamine using these models. For example, line 446 states that Bayesian models outperformed RL models in explaining mouse behavior. However, I don’t think that is completely clear in the results (e.g. in the new analysis in S6, one could say that RLFQ3p actually captures better animals’ choice behavior). This further shows, that behavior and dopamine signaling do not necessarily need to be aligned, as different systems might be differently involved in producing behavior as animals potentially shift from goal-directed to more habitual control.

As for some other minor comments, I think it would be good to also show RFLR in S6, as the authors state that is similar to RLFQ3p. I think it would be useful, the same way it is useful to see in this plot, that both Bayesian and RLCF models show qualitatively similar simulated behavior. Finally, I may have missed it, but supplementary figures S3-6 do not seem to be referenced in the main text.

**Have the authors made all data and (if applicable) computational code underlying the findings in their manuscript fully available?**

Reviewer #1: Yes

Reviewer #2: Yes

PLOS authors have the option to publish the peer review history of their article (what does this mean?). If published, this will include your full peer review and any attached files.

Reviewer #1: No

Reviewer #2: No

**Figure resubmission:**
---

## [Editor Report · Decision Letter 2]

11 Jun 2025

Dear PhD Student Researcher Qü,

We are pleased to inform you that your manuscript 'Nucleus accumbens dopamine release reflects Bayesian inference during instrumental learning' has been provisionally accepted for publication in PLOS Computational Biology.

Best regards,

Tianming Yang

Academic Editor

PLOS Computational Biology

Daniele Marinazzo

Section Editor

PLOS Computational Biology

---

## [Editor Report · Acceptance letter]

PCOMPBIOL-D-24-01557R2

Nucleus accumbens dopamine release reflects Bayesian inference during instrumental learning

Dear Dr Qü,

I am pleased to inform you that your manuscript has been formally accepted for publication in PLOS Computational Biology. Your manuscript is now with our production department and you will be notified of the publication date in due course.

With kind regards,

Zsofia Freund
